# MEMORY-EFFICIENT DIFFERENTIALLY PRIVATE TRAINING WITH GRADIENT RANDOM PROJECTION

## ABSTRACT

Differential privacy (DP) protects sensitive data during neural network training, but standard methods like DP-Adam suffer from high memory overhead due to per-sample gradient clipping, limiting scalability. We introduce DP-GRAPE (Gradient RAndom ProjEction), a DP training method that significantly reduces memory usage while maintaining utility on par with first-order DP approaches. DP-GRAPE is motivated by our finding that privatization flattens the gradient singular value spectrum, making SVD-based projections (as in GaLore Zhao et al. (2024)) unnecessary. Consequently, DP-GRAPE employs three key components: (1) random Gaussian matrices replace SVD-based subspaces, (2) gradients are privatized after projection, and (3) projection is applied during backpropagation. These contributions eliminate the need for costly SVD computations, enable substantial memory savings, and lead to improved utility. Despite operating in lower-dimensional subspaces, our theoretical analysis shows that DP-GRAPE achieves a privacy-utility trade-off comparable to DP-SGD. Our extensive empirical experiments show that DP-GRAPE can reduce the memory footprint of DP training without sacrificing accuracy or training time. In particular, DP-GRAPE reduces memory usage by over 63% when pre-training Vision Transformers and over 70% when fine-tuning RoBERTa-Large as compared to DP-Adam, while achieving similar performance. We further demonstrate that DP-GRAPE scales to fine-tuning large models such as OPT with up to 6.7 billion parameters, a scale at which DP-Adam fails due to memory constraints.

## 1 INTRODUCTION

While deep neural networks have shown enormous performance in Computer Vision and Natural Language Processing, they can be vulnerable to attacks that reveal training instances (Fredrikson et al., 2015; Carlini et al., 2021; Mireshghallah et al., 2022). This poses risks when training on sensitive data, such as personal or medical records. Differential privacy (DP) offers a principled way to protect privacy of individual samples (Dwork et al., 2006b; 2014). In neural network training, methods such as DP-SGD and DP-Adam enforce DP by clipping per-sample gradients and adding Gaussian noise calibrated to the desired privacy level (Abadi et al., 2016).

In addition to privacy, training large models efficiently requires careful memory management. Standard optimizers such as Adam demand substantial memory to store parameters, activations, gradients, and optimizer states—posing a challenge for models with billions of parameters, especially on

Table 1: Comparison of benefits of different memory-saving methods for DP training. We consider a method "high-utility" if it can achieve close to the utility of DP-Adam, see Section 5.2.

| Method | High-Utility | Pre-training | Extra Computation | Low-Memory Optimizer States | Total Number of Steps to Converge |
|---|---|---|---|---|---|
| Bu et al. (2021) | × | ✓ | JL Projections | No | Medium |
| Ghost Clipping | ✓ | ✓ | Extra Backward Pass | No | Medium |
| Book-Keeping | ✓ | ✓ | Extra Partial Gradients | No | Medium |
| Zeroth-order DP | × | × | - | Yes | Large |
| DP-LoRA | × | × | - | Yes | Medium |
| DP-GRAPE | ✓ | ✓ | Low-overhead Random Projection | Yes | Medium |

memory-limited GPUs. To address this, recent work has explored low-rank training methods in non-private settings. LoRA reduces optimizer memory by constraining weight updates to low-rank matrices (Hu et al., 2021), while GaLore projects layer gradients onto SVD-derived subspaces (Zhao et al., 2024). To avoid costly SVD computations at scale, alternatives such as randomized SVD (Pasand & Bashivan, 2024), Gaussian random projections (Hao et al., 2024), and Stiefel manifold methods (He et al., 2024) have been proposed.

When both privacy and memory constraints are present, training becomes even more challenging. DP-SGD and DP-Adam introduce significant overhead by requiring per-sample gradient clipping, with memory usage scaling linearly with model and batch size—often becoming the dominant cost. Ghost clipping reduces this cost by using an extra backward pass to compute sample gradient norms without instantiating full gradients (Lee & Kifer, 2021; Li et al., 2021), but it adds compute overhead and does not reduce optimizer state memory. Book-Keeping (Bu et al., 2023) removes the extra pass using additional tricks, but increases memory usage by storing partial gradients and still offers no savings for optimizer states. As we will see, both of these methods are orthogonal to our approach and can be combined for further memory reduction with our method.

Other strategies have also been explored: Approximating gradient norms via random projections reduces memory but yields poor privacy guarantees at low projection dimensions (Bu et al., 2021). DP-LoRA combines LoRA with differential privacy to reduce trainable parameters, but it is not applicable for pre-training and falls short of DP-Adam in fine-tuning (Yu et al., 2021a) (also see Table 3). Zeroth-order methods such as DP-ZO (Tang et al., 2024) and DPZero (Zhang et al., 2024) are memory-efficient but generally underperform first-order DP methods in NLP fine-tuning and cannot be used for pre-training. In addition, zeroth-order methods require a large number of steps to converge even for non-private training (Malladi et al., 2023; Li et al., 2025). We summarize the trade-offs of these methods in Table 1, including utility, pre-training compatibility, added computation, and optimizer memory usage.

Another related line of work projects gradients onto subspaces obtained from auxiliary non-sensitive data (Yu et al., 2021b; Zhou et al., 2020; Gu et al., 2023). While this approach can improve model utility, it requires having non-sensitive data available, and previous works have not exploited the potential memory benefits of gradient projection to enable scaling to larger models.

To enable memory-efficient DP training for large models without sacrificing utility, we propose DP-GRAPE. It projects sample gradients onto lower-dimensional subspaces using random Gaussian matrices during backpropagation, reducing memory usage for both sample gradients and optimizer states. Our main contributions are:

- We demonstrate that privatization flattens the singular value spectrum of gradients, motivating the use of random projections (as in (Hao et al., 2024)) rather than SVD-based projections (as in (Zhao et al., 2024)).
- We establish that projecting gradients before privatization is critical for achieving high utility and memory efficiency. DP-GRAPE uses the principle of privatizing the *projected gradients* in lower dimensional space to achieve significant memory efficiency gains and improved utility over a naïve approach that privatizes gradients before projection.
- We provide a set of novel analyses of the privacy-utility guarantee for the proposed DP-GRAPE with projection matrices having random but unbounded entries. The theoretical result indicates that DP-GRAPE enjoys the same utility guarantee as DP-SGD.
- Our experiments demonstrate the efficiency and high utility of the proposed algorithm. In particular, compared to DP-Adam, our algorithm achieves similar performance with significantly less memory usage (e.g., 24.4GB compared with 78.1GB when training RoBERTa-Large). Compared to recent memory-efficient DP training methods such as DP-ZO and DP-LoRA, our algorithm can be used for pre-training, while the others are restricted to fine-tuning tasks only. Furthermore, DP-GRAPE requires significantly fewer iterations to converge than zeroth-order DP methods, reducing training time by more than 6 times.

Concurrently with our work, D2P2-SGD (Jiang et al., 2024) introduced random gradient projection in the DP setting, with a focus on tighter theoretical error bounds rather than memory efficiency. A key distinction is that DP-GRAPE projects gradients before privatization, enabling substantial memory savings and improved practical utility, as demonstrated in Section 5.1.

## 2 BACKGROUND

**Notations & Definitions** Throughout the paper, we consider minimizing a loss function $f$ over the dataset $X$ of size $n$ with samples $\{\xi_1, \ldots, \xi_n\}$. We assume the loss is parameterized as a model with $L$ layers, and the $\ell^{\text{th}}$ layer holds trainable parameters as a matrix $W_\ell \in \mathbb{R}^{m_\ell \times n_\ell}$ of size $m_\ell \times n_\ell$. The total number of parameters is $d = \sum_{\ell=1}^{L} m_\ell n_\ell$. The problem we seek to solve is:

$$\min_{\{W_\ell\}_{\ell=1}^{L}} \frac{1}{n} \sum_{i=1}^{n} f(\{W_\ell\}_{\ell=1}^{L}; \xi_i) \tag{1}$$

Without loss of generality, we assume $m_\ell \leq n_\ell$ for all layers. When training the model with iterative methods (e.g., DP-SGD), we denote the total number of training steps as $T$ and index the steps with $(\cdot)^t$, the batch size as $B$, and index the samples with $(\cdot)_i$.

Let $G_{\ell,i}^t = \nabla_{W_\ell^t} f(\{W_\ell^t\}_{\ell=1}^{L}; \xi_i)$ be the gradient for sample $i$ at iteration $t$ for layer $\ell$, and $\{G_{\ell,i}^t\}_{i=1}^{B}$ be the collection of all sample gradients in the batch. We denote the concatenated gradients of all the layers at iteration $t$ for sample $i$ as $G_i^t = \left[ \text{vec}(G_{1,i}^t)^\top \quad \cdots \quad \text{vec}(G_{L,i}^t)^\top \right]^\top$, where $\text{vec}(G_{\ell,i}^t)$ is the vectorized sample gradient for sample $i$ at layer $\ell$ (a column vector of length $m_\ell n_\ell$). The clipping operation used to bound the norm of per-sample gradients is defined as $\text{clip}(G_i^t, C) = \min(1, \frac{C}{\|G_i^t\|_2}) G_i^t$ for sample gradient $G_i^t$ and clipping threshold $C > 0$.

For methods that project gradients, we denote the projection matrix for layer $\ell$ as $P_\ell \in \mathbb{R}^{m_\ell \times r}$, where $r$ is the projection dimension. Let $\mathcal{N}_{s_\ell}(0, \frac{1}{r}) \in \mathbb{R}^{m_\ell \times r}$ be a matrix with entries i.i.d. from a Gaussian distribution with mean 0 and variance $\frac{1}{r}$, generated using seed $s_\ell$. When drawing from a normal distribution without a particular seed (such as when adding noise to gradients), we omit the seed. We denote the projected gradient as $R = P^T G$, and the privatized one as $\tilde{R}$.

**Differential Privacy** Differential privacy ensures that an algorithm's output does not change significantly when a single training sample is removed, protecting individual data. Formally, for neighboring datasets $X$ and $X'$ differing by one sample, a randomized algorithm $\mathcal{A}: D \to O$, where $D$ is the set of all possible datasets and $O$ is the set of all possible outcomes, is $(\varepsilon, \delta)$-DP if (Dwork et al., 2006a):

$$P(\mathcal{A}(X) \in O) \leq e^\varepsilon P(\mathcal{A}(X') \in O) + \delta. \tag{2}$$

A common way to make a function $h: X \to \mathbb{R}^d$ differentially private is to add Gaussian noise scaled to the function's $\ell_2$ sensitivity: $\Delta_2 h := \sup_{X,X'} \|h(X) - h(X')\|_2$ :

**Theorem 2.1** (Dwork et al. (2014)). *Given a function $h: X \mapsto \mathbb{R}^d$ with $\ell_2$ sensitivity $\Delta_2 h$, a dataset $X$, and $\varepsilon, \delta > 0$, the randomized algorithm $\mathcal{A}(X) = h(X) + z$, where $z \sim \mathcal{N}\left(0, \frac{\Delta_2 h}{\varepsilon} \sqrt{2 \log\left(\frac{1.25}{\delta}\right)} \mathbf{I}_d\right)$, is $(\varepsilon, \delta)$-DP.*

**Differentially Private Optimization** To train a differentially private neural network, noise is added to the gradients of each sample during training, rather than to the outputs. Since the gradients' $\ell_2$ sensitivity is often unbounded, they are clipped to a constant $C$ to limit the maximum norm. This gradient clipping, followed by the addition of noise calibrated to the desired privacy level, can be applied to any gradient-based optimization method such as SGD or Adam to achieve $(\varepsilon, \delta)$-DP (Abadi et al., 2016). See Appendix A for the full DP-Adam algorithm. However, gradient privatization can reduce model utility. Additionally, clipping per-sample gradients requires computing individual gradients for each sample (rather than just the gradient averaged over all samples in the batch as in non-private training), increasing memory usage from $d$ to $Bd$. For large models, this forces either a smaller batch size (which may reduce utility) or more gradient accumulation steps (which significantly increases the total training time).

**Memory-Efficient Training with Gradient Projection** One approach to reduce memory usage of training is to project gradients onto lower-dimensional subspaces, allowing optimizer states (e.g., the moment estimates in Adam) to be stored in the subspace. This reduces the memory usage of optimizer states from $\sum_\ell m_\ell n_\ell$ to $r \sum_\ell n_\ell$, significantly saving memory when $r \ll m_\ell$. Various projection methods have been proposed, including using the SVD of layer gradients and random matrices. In GaLore (Zhao et al., 2024), gradients are projected onto subspaces spanned by the top $r$ left singular

vectors from an SVD of the previous layer gradient, with projection matrix $P_\ell^t = U[:, :r]$. FLoRA (Hao et al., 2024) instead uses a Gaussian matrix for $P_\ell^t$, where $(P_\ell^t)_{i,j} \sim \mathcal{N}(0, \frac{1}{r})$. Alternatively, He et al. (2024) use a uniform distribution on the Stiefel manifold to generate $P_\ell^t$.

# 3 METHODOLOGY

Our approach adapts memory-efficient training methods that use gradient projections such as Ga-Lore Zhao et al. (2024) and Flora Hao et al. (2024) to the DP setting. Integrating gradient projection with DP introduces two important design choices: the type of projection (e.g., SVD-based or random), and the order of operations (privatizing gradients before or after projection). We systematically evaluate these choices. Since SVD-based projection derives subspaces from the gradients, to maintain privacy the gradients must be privatized before computing the SVD. However, as shown in Fig. 1, privatization (i.e., clipping and noise addition) flattens the singular value spectrum, destroying any low-rank structure that the SVD aims to capture. This motivates the use of random projections, which can be applied before privatization and is computationally cheaper. For an empirical comparison, we consider an algorithm that privatize gradients before SVD-based projection, which we call naïve DP-GaLore due to its similarity to GaLore, and an algorithm that privatizes after random projection, which is DP-GRAPE.

## 3.1 NAÏVE DP-GALORE

Naïve DP-GaLore computes the SVD of per-sample gradients and applies privatization before projection (see Appendix B for details). However, this approach inherits the high memory cost of storing per-sample gradients of DP-SGD or DP-Adam, offering little improvement over them. In addition, computing SVDs for each layer becomes computationally expensive for large models. Finally, clipping full-dimensional gradients—rather than their projected versions—leads to degraded utility, as discussed in Section 5.1.

## 3.2 OUR ALGORITHM

DP-GRAPE, uses random projection instead of the SVD and applies privatization of gradients after projection. This greatly reduce per-sample gradient memory and lead to significantly better utility (see Section 5.1) as compared to naïve DP-GaLore.

We describe our training method in Algorithm 1, with the projected Adam update in Algorithm 2. For simplicity of presentation, we consider a model with $\ell$ linear layers, each parameterized by a matrix of size $m_\ell \times n_\ell$; nonlinear layers are handled as in DP-Adam (i.e., without projection). Each training iteration consists of a backward pass (steps 2–11), gradient privatization (steps 12–13), and optimizer and weight updates (step 14). To reduce memory, per-sample gradients are projected layer-by-layer during the backward pass (to avoid having the per-sample gradients for each layer being instantiated all at once). The projected gradients are then privatized and used for the projected Adam update.

Our use of random projections rather than the SVD is motivated by empirical observations about the effect of privatization on the singular values. As shown in Fig. 1, while the non-private gradient exhibits some low-rank structure, the combination of clipping and adding noise at levels needed to achieve typically-used levels of DP (e.g., $C = 1.0$ and $\sigma = 0.5, 2.0$) flattens the spectrum of singular values, destroying the low-rank structure. Furthermore, the use of random projections confers additional computational benefits since SVDs for the gradient matrix of each layer do not need to be computed, and projection matrices do not have to be stored for each layer since they can be cheaply generated from a random seed on-the-fly using `torch.randn` (Paszke et al., 2019).

## 3.3 MEMORY REQUIREMENTS

Compared to DP-Adam, DP-GRAPE reduces memory usage by storing projected sample gradients and using lower-dimensional moments in Adam. Table 2 compares the memory requirement of gradients, optimizer states, and projectors for non-private Adam and GaLore, DP-Adam, naïve DP-GaLore, and DP-GRAPE. Notably, DP-GRAPE achieves the largest savings by reducing sample gradient memory from $B \sum_{\ell=1}^{L} m_\ell n_\ell$ (in DP-Adam and naïve DP-GaLore) to $Br \sum_{\ell=1}^{L} n_\ell$, since

Table 2: Memory usage of gradients, optimizer states, and projectors (if used) during training with batch size $B$ for different first-order non-DP and DP methods for an $L$-layer model of sizes $\{m_\ell \times n_\ell\}$. "Gradient" indicates the batch gradient for non-DP methods and all of the sample gradients for the batch for DP method. For GaLore, Naïve DP-GaLore, and DP-GRAPE, $r$ denotes the projection dimension.

| Method | Gradient | Optimizer States | Projectors |
|---|---|---|---|
| Adam (non-private) | $\sum_{\ell=1}^{L} m_\ell n_\ell$ | $2\sum_{\ell=1}^{L} m_\ell n_\ell$ | - |
| GaLore (non-private) | $\sum_{\ell=1}^{L} m_\ell n_\ell$ | $2r\sum_{\ell=1}^{L} n_\ell$ | $r\sum_{\ell=1}^{L} m_\ell$ |
| DP-Adam | $B\sum_{\ell=1}^{L} m_\ell n_\ell$ | $2\sum_{\ell=1}^{L} m_\ell n_\ell$ | - |
| Naïve DP-GaLore | $B\sum_{\ell=1}^{L} m_\ell n_\ell$ | $2r\sum_{\ell=1}^{L} n_\ell$ | $r\sum_{\ell=1}^{L} m_\ell$ |
| DP-GRAPE | $Br\sum_{\ell=1}^{L} n_\ell$ | $2r\sum_{\ell=1}^{L} n_\ell$ | $r\max\{m_\ell\}_{\ell=1}^{L}$ |

$r \ll m_\ell$. Similar to GaLore, it also reduces optimizer memory from $2\sum_{\ell=1}^{L} m_\ell n_\ell$ to $2r\sum_{\ell=1}^{L} n_\ell$. Additionally, using random projections instead of SVD reduces projector memory to $r\max\{m_\ell\}_{\ell=1}^{L}$.

## 4 ANALYZING THE PRIVACY-UTILITY TRADEOFF

We provide the privacy and convergence guarantee of our proposed algorithm below.

**Theorem 4.1** (Informal). *Given a $\Gamma$ lipschtiz and $\lambda$ smooth (potentially non-convex) objective function $f(\cdot; \xi) : \mathbb{R}^d \to \mathbb{R}$ for all $\xi \in X$, for any $0 < \varepsilon \le 2\ln(2/\delta)$ and $\delta \in (0, 1)$, Algorithm 1 is $(\varepsilon, \delta)$-DP if $\sigma = \frac{2C\sqrt{T\log(1/\delta)}}{n\epsilon}$. Moreover, there exist a set of hyper-parameters $\eta, B, C$ such that when $T = \frac{2\sqrt{2}dn\varepsilon}{r\sqrt{L\log(1/\delta)}}$, the output of Algorithm 1, $W_\tau$, for a simple SGD update rule satisfies*

$$\mathbb{E}\left[\|\nabla F(W_\tau)\|^2\right] = \tilde{\mathcal{O}}\left(\frac{\sqrt{Ld\log(1/\delta)}}{n\varepsilon}\right),$$

*where the expectation is taken over all previous sampled batches, random matrices, the additive noise, and sampling of the final parameter vector.*

---

**Algorithm 1** DP-GRAPE

---

**Require:** Dataset $X = \{\xi_1, \ldots, \xi_n\}$, initial weights $\{W_\ell^0\}_{\ell=1}^L$, learning rate $\eta$, subspace dimension $r$, subspace change frequency $F$, batch size $B$, clipping parameter $C$, noise level $\sigma$, total steps $T$
1: **for** $t = 1, 2, \ldots, T$ **do**
2:    **for** $\ell = L, L-1, \ldots, 1$ **do**
3:       $\{G_{\ell,i}^t\}_{i=1}^B \leftarrow \nabla_{W_\ell^t} f(\{W_\ell^t\}_{\ell=1}^L; \{\xi_i\}_{i=1}^B)$
4:       **if** $t \mod F = 0$ **then**
5:          Generate new $s_\ell^t$
6:       **else**
7:          $s_\ell^t \leftarrow s_\ell^{t-1}$
8:       **end if**
9:       $P_\ell^t \leftarrow \mathcal{N}_{s_\ell^t}(0, \frac{1}{r}) \in \mathbb{R}^{m_\ell \times r}$
10:      $R_{\ell,i}^t \leftarrow (P_\ell^t)^\top G_{\ell,i}^t, \quad i = 1, \ldots, B$
11:    **end for**
12:    $R^t \leftarrow \sum_{i=1}^B \text{clip}(R_i^t, C)$
13:    $\tilde{R}^t \leftarrow \frac{1}{B}(R^t + \mathcal{N}(0, C^2\sigma^2 I) \in \mathbb{R}^{r\sum_{\ell=1}^L n_\ell})$
14:    $\{W_\ell^{t+1}\}_{\ell=1}^L = \textbf{Update}(\{W_\ell^t\}_{\ell=1}^L, \{\tilde{R}_\ell^t\}_{\ell=1}^L, \{s_\ell^t\}_\ell^L, \eta)$
15: **end for**
16: Pick $\tau$ uniformly at random from $\{1, 2, \cdots, T\}$.
17: Return $\{W_\ell^\tau\}_{\ell=1}^L$

---

The formal statement of this theorem is Theorem D.10, with assumptions detailed in Appendix D. We follow our result with some remarks that help clarify the implications of our findings and its relevance to our setting.

---

**Algorithm 2** Adam Update with Projected Moments

---

**Require:** Model parameters $\{W_\ell^0\}_{\ell=1}^L$, current projected first-order moments $\{M_\ell^{t-1} \in \mathbb{R}^{r \times n_\ell}\}_{\ell=1}^L$, current projected second-order moments $\{V_\ell^{t-1} \in \mathbb{R}^{r \times n_\ell}\}_{\ell=1}^L$, projector seeds $\{s_\ell^t\}_{\ell=1}^L$, step size $\eta$, decay rates $\beta_1, \beta_2$, iteration $t$, numerical stability constant $\phi$

$\alpha^t \leftarrow \eta \frac{\sqrt{1-\beta_2^t}}{1-\beta_1^t}$

**for** $\ell = 1, 2, \ldots, L$ **do**

$\quad M_\ell^t \leftarrow \beta_1 M_\ell^{t-1} + (1-\beta_1)\tilde{R}_\ell^t$

$\quad V_\ell^t \leftarrow \beta_2 V_\ell^{t-1} + (1-\beta_2)(\tilde{R}_\ell^t)^2$

$\quad P_\ell^t \leftarrow \mathcal{N}_{s_\ell^t} \in \mathbb{R}^{m_\ell \times r}$

$\quad W_\ell^t \leftarrow W_\ell^{t-1} - \alpha^t \cdot P_\ell^t\left(\frac{M_\ell^t}{\sqrt{V_\ell^t}+\phi}\right)$

**end for**

Return $\{W_\ell^t\}_{\ell=1}^L$, $\{M_\ell^t\}_{\ell=1}^L$, $\{V_\ell^t\}_{\ell=1}^L$

---

*Remark 1.* For the case of a constant number of layers $L$, we recover the expected stationarity gap (up to log terms) of DP-SGD, $\mathcal{O}\left(\frac{\sqrt{d\log(1/\delta)}}{n\varepsilon}\right)$ Wang et al. (2017), which is near optimal for the non-convex case (Arora et al., 2023). This is a practically valid setting because the number of layers is constant compared to the number of model parameters $d$. For instance, Llama-3 has $d = 405$B while $L = 128$ (Grattafiori et al., 2024).

*Remark 2.* In addition to the favorable scaling with respect to the number of layers, the dependence of the expected stationarity gap on the number of projections $r$ is minor and captured in the log terms. In fact, the log term decreases as $r$ increases up to a certain point (See Appendix D). However, the increase in random projections significantly lowers (by a factor of $r$) the time required for the algorithm to converge.

These observations provide a broader context for our comparison with Zhang et al. (2024), which offers the first memory-efficient DP optimization guarantee. In the smooth case, their finite-difference step can be interpreted as a special case of our framework with rank-1 ($r = 1$) projection. Our analysis thus generalizes theirs to arbitrary $r \geq 1$. However, their approach samples projection vectors uniformly from the unit sphere, requiring normalization of high-dimensional Gaussian vectors—an operation shown to be a bottleneck in DP optimization due to the cost of computing norms (Bu et al., 2021). This overhead applies not only to gradients but also to random vectors used in projections, particularly when applied block-wise.

This exposes a key challenge: Gaussian-based projections have unbounded norm in the worst case, and clipping exacerbates this by increasing the expected norm of surviving samples. One potential solution involves truncating the Gaussian vector's components to lie within finite symmetric bounds (while adjusting for unbiasedness), ensuring convergence per the framework used in Zhang et al. (2024). However, this approach still incurs similar computational overheads as clipping - whether in terms of memory or runtime - due to the need to truncate each vector entry.

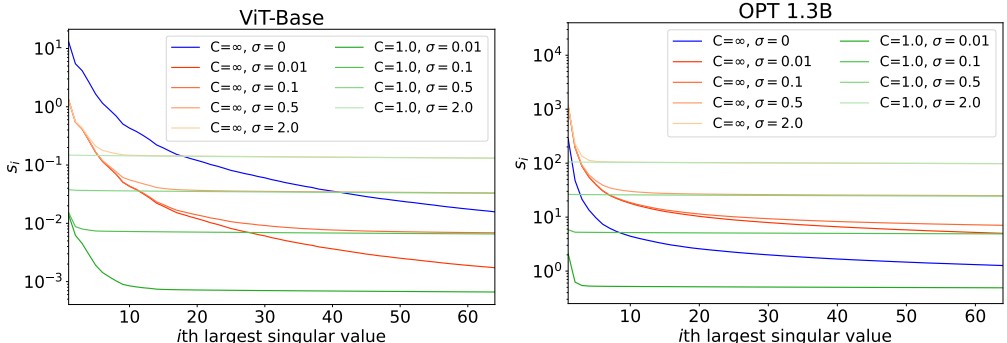

Figure 1: Left: singular values $s_i$ of layer gradient matrices with different clipping parameter $C$ and noise levels $\sigma$, averaged across all layers of ViT-Base during training on CIFAR-10. Right: singular values of gradient matrices for OPT 1.3B during fine-tuning on SST-2. $C = \infty$ indicates no clipping. See Appendix C.1 for details.

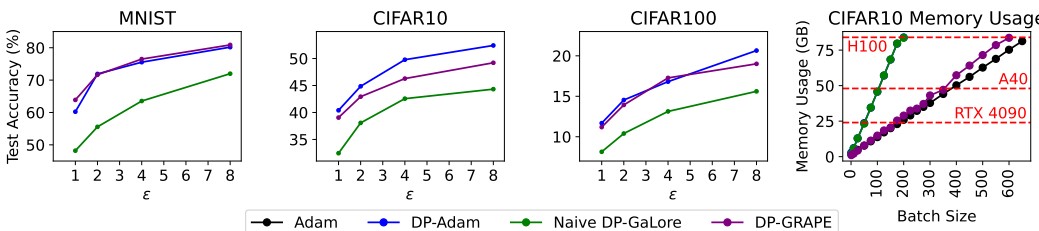

Figure 2: Vision Transformer pre-training results for MNIST, CIFAR-10, and CIFAR100 at different $\varepsilon$ privacy levels, and memory usage for different methods during training with varying batch size, with non-private Adam for comparison. See Appendix C.1 for detailed results in table form and experiment setup.

Through careful analysis, we demonstrate that sampling vectors with unbounded worst-case norms does not adversely affect convergence due to the exponential decay of tail probabilities, even in expectation. This insight allows us to exploit the efficient generation of standard normal vectors without requiring additional processing, thus addressing the computational challenges while maintaining theoretical guarantees.

## 5 EXPERIMENTS

We conduct experiments across three tasks to evaluate the performance of DP-GRAPE and compare it against other DP methods: 1) **Pre-training** - training a VIT-base model from scratch on image classification tasks; 2) **Fine-tuning** - fine-tuning a RoBERTa-Large model on text classification tasks; 3) **Scalability** - demonstrating the scalability of DP-GRAPE by successfully fine-tuning OPT models ranging from 1B to 6.7B parameters. Detailed experiment settings are provided in Appendix C.

### 5.1 VISION TRANSFORMER TRAINING

To evaluate the effectiveness of DP-GRAPE on pretraining tasks, we train Vision Transformer models (base model, 85M parameters) from scratch on MNIST (Deng, 2012), CIFAR10, and CIFAR100 (Krizhevsky et al., 2009). To compare the performance and memory usage, we also train models using DP-Adam and naïve DP-GaLore, as discussed in Section 3.1 (for pretraining, DP-LoRA and DP-Zero are not typically used, so we do not include comparisons). For all methods, we select the best clipping threshold and learning rate from a grid search, and then use those hyperparameters to train models for each method at $\varepsilon = 1, 2, 4, 8$ privacy levels, with $\delta = \frac{1}{n}$. We do not use any additional public data or data augmentation. We discuss the experimental setup in detail in Appendix C.1.

**Utility:** Figure 2 shows both the final test accuracies for the different methods and datasets at varied privacy levels and the memory usage for each method across different batch sizes, with non-private Adam for comparison. See Appendix C.1 for a full table of results. We find that while the naïve DP-GaLore approach performs significantly worse than DP-Adam, with an average decrease in accuracy across the different privacy levels of $12.1\%$ on MNIST, $7.8\%$ on CIFAR-10, and $4.1\%$ on CIFAR-100, DP-GRAPE achieves comparable performance to DP-Adam. Averaged across privacy levels, DP-GRAPE improves over DP-Adam by $1.3\%$ on MNIST, decreases by $2.5\%$ on CIFAR-10, and decreases by $0.6\%$ on CIFAR-100. While the accuracy of the models pre-trained with DP are relatively low, there are various strategies for improving performance, including data augmentation techniques (De et al., 2022; Bao et al., 2023) and training with limited public data (Bu et al., 2024). However, in order to directly compare DP-Adam, the naïve DP-GaLore, and DP-GRAPE, we do not integrate these techniques. In Appendix C.1, we also include results for fine-tuning on CIFAR-10 and CIFAR-100, where we find that DP-GRAPE achieves similar accuracy as DP-Adam on CIFAR-10 and achieves significantly better accuracy on CIFAR-100.

**Memory Usage:** While achieving nearly the same accuracy as DP-Adam, DP-GRAPE uses significantly less memory during training. When training on CIFAR-10 using a 24GB GPU, DP-GRAPE allows a maximum batch size of around 165, while DP-Adam only allows for a maximum batch size of about 50. Consequently, given a fixed memory budget, DP-GRAPE achieves a $25\%$ increase in throughput compared with DP-Adam, as shown in Table 7 in Appendix C.1.

Table 3: Mean and standard error of final test accuracy over three different seeds for few-shot ($k = 512$) fine-tuning of RoBERTa-Large on different datasets, for different DP and non-private methods. The best DP result for each privacy level and dataset is in bold. See Appendix C.2 for experiment details.

| Task | SST-2 | SST-5 | SNLI | MNLI | RTE | TREC |
|------|-------|-------|------|------|-----|------|
| AdamW (non-private) | $93.1 \pm 0.3$ | $56.6 \pm 0.3$ | $86.4 \pm 0.8$ | $81.4 \pm 0.9$ | $83.6 \pm 1.6$ | $95.9 \pm 0.2$ |
| DP-Adam ($\varepsilon = 6$) | $91.6 \pm 1.2$ | $49.0 \pm 0.3$ | $81.5 \pm 1.4$ | $76.3 \pm 0.9$ | $\mathbf{77.3 \pm 1.1}$ | $89.9 \pm 0.8$ |
| DP-Adam ($\varepsilon = 2$) | $90.5 \pm 1.5$ | $\mathbf{47.5 \pm 0.5}$ | $74.6 \pm 1.0$ | $\mathbf{70.3 \pm 0.8}$ | $\mathbf{72.8 \pm 0.9}$ | $85.0 \pm 0.5$ |
| LoRA (non-private) | $93.3 \pm 0.4$ | $55.3 \pm 1.0$ | $85.9 \pm 0.7$ | $82.2 \pm 0.7$ | $84.2 \pm 0.4$ | $94.6 \pm 0.4$ |
| DP-LoRA ($\varepsilon = 6$) | $91.0 \pm 1.3$ | $48.8 \pm 0.5$ | $81.0 \pm 1.5$ | $72.8 \pm 1.8$ | $74.7 \pm 1.3$ | $89.2 \pm 0.8$ |
| DP-LoRA ($\varepsilon = 2$) | $90.2 \pm 1.2$ | $47.1 \pm 0.4$ | $74.7 \pm 1.6$ | $65.7 \pm 0.9$ | $69.2 \pm 1.1$ | $83.2 \pm 2.3$ |
| MeZO (non-private) | $92.5 \pm 0.3$ | $50.8 \pm 0.8$ | $80.4 \pm 0.6$ | $69.2 \pm 0.3$ | $72.8 \pm 1.0$ | $88.9 \pm 0.1$ |
| DPZero ($\varepsilon = 6$) | $92.2 \pm 0.3$ | $\mathbf{49.3 \pm 0.6}$ | $77.8 \pm 1.0$ | $67.4 \pm 0.3$ | $71.9 \pm 0.9$ | $87.6 \pm 0.9$ |
| DPZero ($\varepsilon = 2$) | $91.8 \pm 0.1$ | $47.1 \pm 0.9$ | $73.6 \pm 0.9$ | $62.7 \pm 0.9$ | $70.4 \pm 0.7$ | $82.0 \pm 1.6$ |
| DP-GRAPE ($\varepsilon = 6$) | $\mathbf{93.3 \pm 0.4}$ | $49.1 \pm 0.1$ | $\mathbf{83.5 \pm 0.4}$ | $\mathbf{76.7 \pm 0.4}$ | $76.4 \pm 0.8$ | $\mathbf{92.7 \pm 1.0}$ |
| DP-GRAPE ($\varepsilon = 2$) | $\mathbf{92.6 \pm 0.5}$ | $44.5 \pm 0.4$ | $\mathbf{79.6 \pm 0.4}$ | $68.8 \pm 1.2$ | $\mathbf{72.8 \pm 0.9}$ | $\mathbf{88.1 \pm 2.2}$ |
| Zero-Shot | $79.0$ | $35.5$ | $50.2$ | $48.8$ | $51.4$ | $32.0$ |

## 5.2 RoBERTa Fine-Tuning

We evaluate DP-GRAPE on NLP fine-tuning tasks by fine-tuning RoBERTa-Large models (355M parameters) (Liu, 2019) from Hugging Face[1] on different sentence classification tasks. Our experimental setup is the same as in Malladi et al. (2023) and Zhang et al. (2024): we use a few-shot setting with 512 samples for each class in all of the datasets. We fine-tune models with DP-GRAPE at both $(\varepsilon = 2, \delta = 1e - 5)$ and $(\varepsilon = 6, \delta = 1e - 5)$ privacy. We detail our hyperparameter selection in Appendix C.

**Utility:** Table 3 shows the results for DP-GRAPE along with other DP and non-private baselines. DP-GRAPE achieves a higher average test accuracy at both $\varepsilon = 2$ and $\varepsilon = 6$ than DPZero on all but one of the six datasets we tested on. On average, across the six datasets and two privacy levels, DP-GRAPE improves upon the test accuracy of DPZero by $3.7\%$. Furthermore, DP-GRAPE is competitive with DP-Adam, improving on it by an average of $1.0\%$ over the different datasets and privacy levels.

**Memory Usage:** In addition to achieving comparable utility, DP-GRAPE uses less memory than DP-Adam, which we illustrate in Fig. 3. When fine-tuning on SST-2 with a batch size of 40, DP-Adam uses 78.1 GB of memory, while DP-GRAPE uses only 24.4 GB of memory, a $68.7\%$ reduction. While DPZero is very memory efficient, it takes about 10 times as many iterations to converge, which we illustrate in Fig. 5 in Appendix C.2. Consequently, for the same experimental setup we use to generate the test results, DPZero takes almost 3 times as long as DP-GRAPE to run (Table 10). As compared to the Vision Transformer pre-training, there is a bigger gap in memory usage between DP-GRAPE and non-private Adam, which is due to gradients from the embedding layers and language modeling head not being projected. In Appendix C.2 we also show the effect of varying the subspace dimension $r$ on the total memory usage of DP-GRAPE.

## 5.3 OPT Fine-Tuning

To assess the scalability of DP-GRAPE to larger models, we use it to fine-tune OPT models (Zhang et al., 2022) with 1.3B, 2.7B, and 6.7B parameters from Hugging Face[2] on both classification and generation tasks. We use the same setup as in Malladi et al. (2023) and Zhang et al. (2024). For details, see Appendix C.3. Table 4 and Table 12 show the results for DP-GRAPE and baselines for the classification and generation tasks, respectively.

**Scaling to Larger Models:** While fine-tuning all parameters of the 6.7B model on any of the datasets with DP-Adam or even non-private Adam on a single 80GB GPU is impossible, DP-GRAPE scales

---

[1]Link to the checkpoint: `https://huggingface.co/FacebookAI/roberta-large`

[2]Link to the checkpoints: `https://huggingface.co/collections/facebook/opt-66ed00e15599f02966818844`

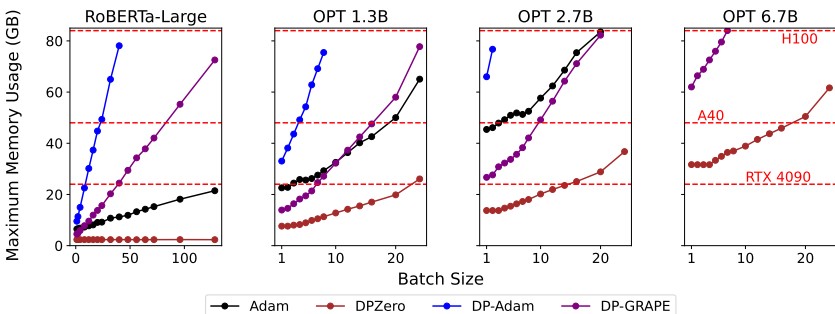

Figure 3: Maximum memory usage for fine-tuning RoBERTa-Large on SST-2 and OPT models on SQuAD using Adam, DP-Adam, DPZero, and DP-GRAPE with varying batch size. See Appendix C.2 and Appendix C.3 for details.

Table 4: Mean and standard error of final test accuracy over three different seeds for few-shot ($k = 1000$) fine-tuning of OPT models on SST-2 and BoolQ classification tasks, for different DP and non-private methods. The best DP result for each privacy level and dataset is in bold. OOM indicates out-of-memory on an 80GB GPU with a batch size of 1 and gradient accumulation. See Appendix C.3 for experiment details.

| Model | OPT-1.3B | | OPT-2.7B | | OPT-6.7B | |
| Task | SST-2 | BoolQ | SST-2 | BoolQ | SST-2 | BoolQ |
| --- | --- | --- | --- | --- | --- | --- |
| MeZO (non-private) | $88.2 \pm 0.9$ | $63.2 \pm 0.8$ | $91.9 \pm 0.5$ | $65.3 \pm 1.3$ | $93.0 \pm 0.2$ | $67.4 \pm 2.3$ |
| DPZero ($\varepsilon = 6$) | $88.2 \pm 1.1$ | $62.4 \pm 0.8$ | $91.5 \pm 1.7$ | $\mathbf{65.4 \pm 1.6}$ | $92.6 \pm 0.7$ | $\mathbf{66.8 \pm 1.6}$ |
| DPZero ($\varepsilon = 2$) | $86.8 \pm 1.7$ | $61.6 \pm 1.1$ | $90.5 \pm 0.9$ | $\mathbf{63.7 \pm 0.7}$ | $90.6 \pm 1.3$ | $\mathbf{63.7 \pm 0.7}$ |
| DP-Adam ($\varepsilon = 6$) | $\mathbf{91.2 \pm 0.2}$ | $62.3 \pm 0.4$ | $\mathbf{93.2 \pm 0.1}$ | $62.9 \pm 0.4$ | OOM | OOM |
| DP-Adam ($\varepsilon = 2$) | $85.9 \pm 0.8$ | $59.8 \pm 0.7$ | $92.4 \pm 0.04$ | $62.5 \pm 0.3$ | OOM | OOM |
| DP-GRAPE ($\varepsilon = 6$) | $90.8 \pm 0.5$ | $\mathbf{62.5 \pm 0.3}$ | $93.0 \pm 0.5$ | $62.7 \pm 0.6$ | $\mathbf{94.2 \pm 0.3}$ | $63.4 \pm 0.8$ |
| DP-GRAPE ($\varepsilon = 2$) | $\mathbf{90.5 \pm 0.3}$ | $\mathbf{62.0 \pm 0.4}$ | $\mathbf{92.5 \pm 0.3}$ | $61.6 \pm 1.3$ | $\mathbf{91.2 \pm 0.2}$ | $63.7 \pm 0.2$ |
| Zero-Shot | 53.6 | 45.3 | 56.3 | 47.7 | 61.2 | 59.4 |

to the 6.7B model. We show the memory usage for DP-GRAPE and comparison methods with different model sizes in Fig. 3. DP-GRAPE achieves better utility than DPZero on 18 of the 24 total combinations of model sizes, datasets, and privacy levels, and better utility than DP-Adam on 9 of 16 total combinations (excluding the 6.7B model). Furthermore, because DP-GRAPE requires 10 times fewer iterations to converge as DP-Zero, it reduces the total fine-tuning time for the 6.7B model on SQuAD using a single H100 GPU by more than 6 times, as shown in Table 13.

## 6 CONCLUSION

DP-GRAPE is a memory-efficient DP training method that achieves utility comparable to standard first-order DP methods while significantly reducing the memory usage of per-sample gradients and optimizer states. We experimentally verify DP-GRAPE on a variety of tasks including pre-training Vision Transformers, fine-tuning RoBERTa-Large on text classification tasks, and fine-tuning OPT models of different sizes on text classification and generation tasks. For the OPT fine-tuning, DP-GRAPE is able to scale to the model size of 6.7B, while DP-Adam is infeasible even with a batch size of 1. Theoretically, DP-GRAPE achieves a similar privacy-utility trade-off to DP-SGD. Techniques like Ghost Clipping (Li et al., 2021) and Book-Keeping (Bu et al., 2023), which reduce memory by avoiding per-sample gradient instantiation, can be combined with DP-GRAPE to further lower memory usage at the cost of increased per-iteration time. Since they only affect gradient computation and not the optimization algorithm, the utility-privacy trade-off of DP-GRAPE remains unchanged. Other potential improvements to DP-GRAPE include adapting the projection dimension across layers based on a selection criterion and utilizing non-Gaussian projection matrices, such as those sampled from a Stiefel manifold. Overall, by reducing resource requirements, DP-GRAPE empowers resource-constrained communities and institutions to build and leverage large models while ensuring data privacy, democratizing access to privacy-preserving AI.

REPRODUCIBILITY STATEMENT

We have provided the source code for DP-GRAPE and all of the experiments from this manuscript in the supplementary materials. In Appendix C.1, Appendix C.2, and Appendix C.3, we provide detailed explanations of how each experiment was run, the hyperparameters we used for each experiment, and the procedures we used to select the hyperparameters. Furthermore, the code contains scripts that can be used to reproduce each experiment. We describe the computational resources we used in Appendix C.5. All datasets we used are publicly available.

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

## A  DP-Adam

Here, we detail the standard DP-Adam algorithm (with flat clipping) using our notation.

## B  GaLore and Naïve DP-GaLore

Here, we detail the GaLore algorithm introduced by Zhao et al. (2024) and the naïve version of DP-GaLore that we discuss in Section 3.1. For naïve DP-GaLore, gradients are privatized prior to projection and before the SVD updates, so that the subspaces obtained from the SVD can be used in subsequent iterations with no privacy loss.

**Algorithm 3** DP-Adam

**Require:** Dataset $X = \{\xi_1, \ldots, \xi_n\}$, model parameters $\{W_\ell^0\}_{\ell=1}^L$, learning rate $\eta$, decay rates $\beta_1, \beta_2$, batch size $B$, total iterations $T$
1: **for** $t = 1, 2, \ldots, T$ **do**
2:    **for** $\ell = L, L-1, \ldots, 1$ **do**
3:      $\{G_{\ell,i}^t\}_{i=1}^B \leftarrow \nabla_{W_\ell^t} f(\{W_\ell^t\}_{\ell=1}^L; \{\xi_i\}_{i=1}^B)$
4:    **end for**
5:    $\tilde{G}^t \leftarrow \frac{1}{B}(\sum_{i=1}^B \text{clip}(G_i^t, C) + \mathcal{N}(0, C^2\sigma^2 I) \in \mathbb{R}^d)$
6:    $\alpha^t \leftarrow \eta \frac{\sqrt{1-\beta_2^t}}{1-\beta_1^t}$
7:    **for** $\ell = 1, 2, \ldots, L$ **do**
8:      $M_\ell^t \leftarrow \beta_1 M_\ell^{t-1} + (1-\beta_1)\tilde{G}_\ell^t$
9:      $V_\ell^t \leftarrow \beta_2 V_\ell^{t-1} + (1-\beta_2)(\tilde{G}_\ell^t)^2$
10:     $W_\ell^t \leftarrow W_\ell^{t-1} - \alpha^t \cdot \left(\frac{M_\ell^t}{\sqrt{V_\ell^t}+\phi}\right)$
11:   **end for**
12: **end for**
13: Return $\{W_\ell^T\}_{\ell=1}^L$

**Algorithm 4** GaLore

**Require:** Dataset $X = \{\xi_1, \ldots, \xi_n\}$, model parameters $\{W_\ell^0\}_{\ell=1}^L$, learning rate $\eta$, subspace dimension $r$, subspace change frequency $F$, batch size $B$, total iterations $T$
1: **for** $t = 1, 2, \ldots, T$ **do**
2:    **for** $\ell = L, L-1, \ldots, 1$ **do**
3:      $G_\ell^t \leftarrow \nabla_{W_\ell^t} f(\{W_\ell^t\}_{\ell=1}^L; \{\xi_i\}_{i=1}^B)$
4:      **if** $t \mod F = 0$ **then**
5:        $U, S, V \leftarrow \text{SVD}(G_\ell^t)$
6:        $P_\ell^t \leftarrow U[:, :r]$
7:      **else**
8:        $P_\ell^t \leftarrow P_\ell^{t-1}$
9:      **end if**
10:     $R_\ell^t \leftarrow (P_\ell^t)^\top G_\ell^t$
11:   **end for**
12:   $\{W_\ell^{t+1}\}_{\ell=1}^L = \mathbf{Update}(\{W_\ell^t\}_{\ell=1}^L, \{R_\ell^t\}_{\ell=1}^L, \{P_\ell^t\}_{\ell=1}^L, \eta)$
13: **end for**
14: Return $\{W_\ell^T\}_{\ell=1}^L$

---

**Algorithm 5** Naïve DP-GaLore

---

**Require:** Dataset $X = \{\xi_1, \ldots, \xi_n\}$, model parameters $\{W_\ell^0\}_{\ell=1}^L$, learning rate $\eta$, subspace dimension $r$, subspace change frequency $F$, batch size $B$, clipping parameter $C$, noise level $\sigma$, total iterations $T$

1: **for** $t = 1, 2, \ldots, T$ **do**
2:     **for** $\ell = L, L-1, \ldots, 1$ **do**
3:         $\{G_{\ell,i}^t\}_{i=1}^B \leftarrow \nabla_{W_\ell^t} f(\{W_\ell^t\}_{\ell=1}^L; \{\xi_i\}_{i=1}^B)$
4:     **end for**
5:     $\tilde{G}^t \leftarrow \frac{1}{B}(\sum_{i=1}^B \text{clip}(G_i^t, C) + \mathcal{N}(0, C^2\sigma^2 I) \in \mathbb{R}^d)$
6:     **for** $\ell = L, L-1, \ldots, 1$ **do**
7:         **if** $t \mod F = 0$ **then**
8:             $U, S, V \leftarrow \text{SVD}(\tilde{G}_\ell^t)$
9:             $P_\ell^t \leftarrow U[:, : r]$
10:         **else**
11:             $P_\ell^t \leftarrow P_\ell^{t-1}$
12:         **end if**
13:         $\tilde{R}_\ell^t \leftarrow (P_\ell^t)^\top \tilde{G}_\ell^t$
14:     **end for**
15:     $\{W_\ell^{t+1}\}_{\ell=1}^L = \textbf{Update}(\{W_\ell^t\}_{\ell=1}^L, \{\tilde{R}_\ell^t\}_{\ell=1}^L, \{P_\ell^t\}_{\ell=1}^L, \eta)$
16: **end for**
17: Return $\{W_\ell^T\}_{\ell=1}^L$

---

## C  EXPERIMENT DETAILS

### C.1  VISION TRANSFORMER TRAINING

We use the same grid search for all three methods, selecting the clipping parameter $C$ from $\{0.1, 1, 10\}$ and the learning rate from $\{1e-4, 5e-4, 1e-3, 5e-3\}$. For the grid-search, we split the training set of each dataset randomly into $80\%$ training and $20\%$ testing data and select the combination of $C$ and learning rate which achieves the highest validation accuracy during training. We use a privacy level of $(\varepsilon = 2, \delta = \frac{1}{n})$ for the grid search. The best hyperparameters are then used to train models on the entire training set at the different privacy levels $\varepsilon = 1, 2, 4, 8$, which are evaluated on the original testing set. Table 5 lists the selected $C$ and learning rate for each method and dataset. For all experiments (both the grid search and final training runs), we use a total batch size of 1000 (which is achieved through gradient accumulation) and train for 60 epochs. In Table 6 we list the best test accuracy during training for each method on the different datasets and privacy levels.

Table 5: Clipping parameter $C$ and learning rate selected from grid search for each method and dataset for Vision Transformer pre-training.

| Method | $C$ | | | Learning Rate | | |
|---|---|---|---|---|---|---|
| | MNIST | CIFAR-10 | CIFAR-100 | MNIST | CIFAR-10 | CIFAR-100 |
| DP-Adam | 10.0 | 0.1 | 1.0 | 1e-3 | 1e-3 | 5e-4 |
| Naïve DP-GaLore | 1.0 | 1.0 | 0.1 | 5e-4 | 5e-4 | 1e-3 |
| DP-GRAPE | 0.1 | 1.0 | 10.0 | 5e-3 | 1e-3 | 1e-3 |

For the memory experiment with results shown in Fig. 2, we train for 5 steps and record the maximum memory reserved by PyTorch (Paszke et al., 2019) using the `torch.cuda.max_memory_reserved()` function, for a range of batch sizes. For all sizes, we use a gradient accumulation step so that accumulated gradients are included in the memory accounting.

To generate the timing results shown in Table 7, we train each method for 1 epoch and then extrapolate the time taken to complete 60 epochs of training. We match the setup we use to generate the results in Fig. 2, with a total batch size of 1000 that is achieved by gradient accumulation. For Adam and DP-GRAPE we use a physical batch size of 500 (which uses 63.0GB and 70.6GB of memory,

Table 6: Vision Transformer pretraining results for MNIST, CIFAR-10, and CIFAR100 at different privacy levels (best test accuracy during training).

| Task | MNIST | CIFAR-10 | CIFAR-100 |
|------|-------|----------|-----------|
| DP-Adam ($\varepsilon = 1$) | 60.2 | 40.4 | 11.7 |
| DP-Adam ($\varepsilon = 2$) | 71.9 | 44.9 | 14.5 |
| DP-Adam ($\varepsilon = 4$) | 75.5 | 49.8 | 16.8 |
| DP-Adam ($\varepsilon = 8$) | 80.2 | 52.4 | 20.7 |
| Naïve DP-GaLore ($\varepsilon = 1$) | 48.2 | 32.5 | 8.1 |
| Naïve DP-GaLore ($\varepsilon = 2$) | 55.6 | 38.1 | 10.4 |
| Naïve DP-GaLore ($\varepsilon = 4$) | 63.5 | 42.6 | 13.1 |
| Naïve DP-GaLore ($\varepsilon = 8$) | 72.0 | 44.3 | 15.6 |
| DP-GRAPE ($\varepsilon = 1$) | 63.9 | 39.1 | 11.2 |
| DP-GRAPE ($\varepsilon = 2$) | 71.7 | 42.9 | 13.9 |
| DP-GRAPE ($\varepsilon = 4$) | 76.5 | 46.3 | 17.3 |
| DP-GRAPE ($\varepsilon = 8$) | 80.9 | 49.2 | 19.0 |

Table 7: Number of samples processed per second during training and total train time for Vision Transformer on CIFAR-10 (using 1 H100 GPU).

| Method | Throughput (Samples/s) | Total Training Time (hours) |
|--------|------------------------|------------------------------|
| Adam (non-private) | 379 | 2.2 |
| DP-Adam | 219 | 3.8 |
| Naïve DP-GaLore | 217 | 3.8 |
| DP-GRAPE | 273 | 3.1 |

respectively), and for DP-Adam and Naïve DP-GaLore we use a physical batch size of 200 (which uses 83.9 and 83.7GB of memory, respectively).

The memory and timing experiments were conducted on a single H100 GPU.

To create the plot of singular values shown in Fig. 1, we record the top $64$ (corresponding to the projection dimension we use for all Vision Transformer experiments) singular values for each layer during the first step of training with a batch size of 1000, with possible clipping and different noise levels $\sigma$ applied to the gradients prior to computing the SVD.

We also fine-tune a pretrained checkpoint of ViT-Base on CIFAR10 and CIFAR100 [3]. For these experiments, we again use a grid search over the training set for all three methods to select the clipping parameter $C$ from $\{0.1, 1, 10\}$ and the learning rate from $\{1e-5, 5e-5, 1e-4, 5e-4\}$. Using the best hyperparameters for each method, we then fine-tune on the entire training set at the different privacy level $\varepsilon = 1, 2, 4, 8$ and evaluate on the original testing set. Table 8 lists the selected $C$ and learning rate for each method and dataset. For both the hyperparameter search and the final fine-tuning, we use a total batch size of 1000 and train for 20 epochs. Table 9 lists the final results (best test accuracy) for each method and dataset at different privacy levels.

Table 8: Clipping parameter $C$ and learning rate selected from grid search for each method and dataset for Vision Transformer fine-tuning.

| Method | $C$ | | Learning Rate | |
|--------|-----|-----|---------------|-----|
| | CIFAR-10 | CIFAR-100 | CIFAR-10 | CIFAR-100 |
| DP-Adam | 10.0 | 10.0 | 1e-4 | 5e-4 |
| Naïve DP-GaLore | 10.0 | 1.0 | 5e-4 | 5e-4 |
| DP-GRAPE | 0.1 | 1.0 | 5e-4 | 5e-4 |

---

[3]Link to the checkpoint: `https://huggingface.co/google/vit-base-patch16-224`

Table 9: Vision Transformer fine-tuning results for CIFAR-10, and CIFAR100 at different privacy levels (best test accuracy during training).

| Task | CIFAR-10 | CIFAR-100 |
|---|---|---|
| DP-Adam ($\varepsilon = 1$) | 97.2 | 49.7 |
| DP-Adam ($\varepsilon = 2$) | 97.5 | 70.6 |
| DP-Adam ($\varepsilon = 4$) | 98.1 | 77.3 |
| DP-Adam ($\varepsilon = 8$) | 98.2 | 80.8 |
| Naïve DP-GaLore ($\varepsilon = 1$) | 96.4 | 65.6 |
| Naïve DP-GaLore ($\varepsilon = 2$) | 97.0 | 78.8 |
| Naïve DP-GaLore ($\varepsilon = 4$) | 97.5 | 83.8 |
| Naïve DP-GaLore ($\varepsilon = 8$) | 97.7 | 85.5 |
| DP-GRAPE ($\varepsilon = 1$) | 97.0 | 81.4 |
| DP-GRAPE ($\varepsilon = 2$) | 97.9 | 85.4 |
| DP-GRAPE ($\varepsilon = 4$) | 97.8 | 86.9 |
| DP-GRAPE ($\varepsilon = 8$) | 98.2 | 88.1 |

## C.2 RoBERTa Fine-Tuning

We follow the same experimental setup and build off of the same codebase as used by Zhang et al. (2024) and Malladi et al. (2023) to fine-tune RoBERTa-Large (Liu, 2019) on datasets that cover sentiment analysis (SST-2, SST-5), natural language inference (SNLI, MNLI, RTE), and topic classification (TREC). For all datasets, we use a few-shot setting with 512 samples per class, and 1000 total test samples. We first complete a grid search to find reasonable values for the projection dimension $r$, the projection update frequency $F$, the learning rate $\eta$, the DP clipping parameter $C$, and the total number of training steps $T$, for the SST-2 and MNLI datasets, evaluating on the development set for seed 100. Based on these experiments, we select $r = 16$, $F = 100$, $\eta = 1\mathrm{e}-4$, and $T = 1000$. Using these values, we run a search for only the clipping parameter for the remaining datasets, selecting from $\{0.1, 0.5, 1.0, 5.0, 10.0, 20.0\}$ (again evaluating on the development set for seed 100). The grid search and clipping parameter search are done with ($\varepsilon = 6, \delta = 1\mathrm{e}-5$) privacy. The best $C$ value from this search is $0.5$ for SST-2, $20.0$ for SST-5, $0.1$ for SNLI, $10.0$ for MNLI, $0.5$ for RTEm and $0.5$ for TREC. Using the best $C$ value, we run the final results for each dataset on the seeds 13, 21, and 42 (which contain different samplings of the full datasets), at both ($\varepsilon = 2, \delta = 1\mathrm{e}-5$) and ($\varepsilon = 6, \delta = 1\mathrm{e}-5$) privacy for each, and record the average final test accuracy over the 3 seeds for each dataset and privacy level. We train with a batch size of $64$ for all experiments, which may be achieved using gradient accumulation. The results for AdamW (non-private), DP-Adam, LoRA (non-private), DP-LoRA, MeZO (non-private), and DPZero come from Zhang et al. (2024).

For the memory experiment with results shown in in Fig. 3, we train for 30 steps and record the maximum memory reserved by PyTorch (Paszke et al., 2019) using the `torch.cuda.max_memory_reserved()` function, for a range of batch sizes. For all sizes, we use a gradient accumulation step for the first-order methods so that accumulated gradients are included in the memory accounting. We also repeat the memory experiment for DP-GRAPE with varying subspace dimension $r$, with results shown in Fig. 4.

To get the timing results shown in Table 10, we time how long the RoBERTa fine-tuning takes on SST-2 takes using an H100 GPU for DP-Adam, DPZero, and DP-GRAPE, using the same experimental setup as we use to get the final results for different. The total batch size is set to 64. For DP-Adam, a batch size of 64 does not fit, so we use a physical batch size of 32 and gradient accumulation. For all methods, we fine-tune for 50 steps. The total train time for each method is inferred from the time for 50 steps, assuming 1000 total steps for DP-Adam and DP-GRAPE and 10000 total steps for DPZero.

The memory and timing experiments were conducted on a single H100 GPU.

To generate the convergence plot shown in Fig. 5, we fine-tune RoBERTa-Large on SST-2 using DP-GRAPE and DPZero (Zhang et al., 2024), and measure the development set accuracy every 50 steps. For DP-GRAPE, we exactly match the experimental setup used to generate the results in Table 3. For DPZero, we use the same setup and implementation as given in the official GitHub implementation.

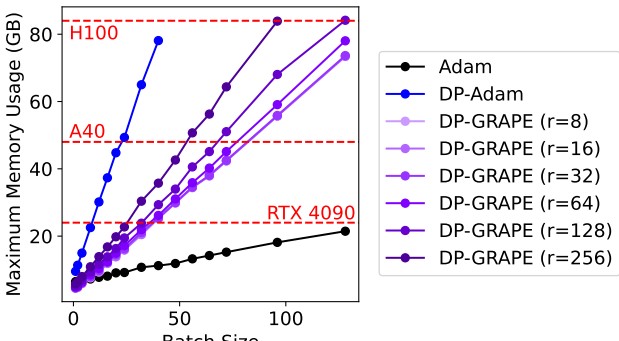

Figure 4: Maximum memory usage for fine-tuning RoBERTa-Large with DP-GRAPE using different subspace dimensions $r$, with comparisons to Adam, DP-Adam, and DPZero.

Table 10: Throughput and total training time for fine-tuning on SST-2 with RoBERTa-Large with a total batch size of 64 on an H100 GPU. Total training time is based on 1000 total steps for DP-Adam and DP-GRAPE (the same total number of steps we use to generate the results in Table 3) and 10000 total steps for DPZero, (the number of steps reported to generate the final results in Zhang et al. (2024).

| Method | Throughput (Samples/s) | Total Train Time (hours) |
|---|---|---|
| DP-Adam | 71.7 | 0.6 |
| DPZero | 268.1 | 1.7 |
| DP-GRAPE | 75.9 | 0.6 |

## C.3 OPT FINE-TUNING

For the OPT experiments, we also follow the same experimental setup and build off the same codebase as used by Zhang et al. (2024) and Malladi et al. (2023) to fine-tune OPT models with 1.3B, 2.7B, and 6.7B parameters on SST-2, BoolQ, and SQuAD, and DROP. We use a few-shot (1000 total training samples) setting for all datasets, and 1000 total samples for testing. Due to the increased computational requirements needing for fine-tuning as compared to the RoBERTa models, we search only for the best clipping parameter $C$ from the choices $\{0.1, 1.0, 5.0, 20.0\}$ for each model and dataset. Table 11 shows the $C$ we select for each model size and dataset for DP-Adam, and DP-GRAPE. For DP-GRAPE, we set $\eta = 1e - 4$, $F = 100$, $T = 2000$, and $r = 16, 32, 64$ for the 1.3B, 2.7B, and 6.7B models, respectively. For DP-Adam, we set $\eta = 1e - 5$ and $T = 3000$ after noting that a smaller learning rate and increased number of training steps is more stable. We train with a batch size of $8$ for all experiments, which may be achieved by gradient accumulation.

For the memory experiment, we train for 30 steps and record the maximum memory reserved by PyTorch (Paszke et al., 2019) using the `torch.cuda.max_memory_reserved()` function. For all sizes, we use a gradient accumulation step for the first-order methods so that accumulated gradients are included in the memory accounting.

The memory and timing experiments were conducted on a single H100 GPU.

For the timing experiment, we use the same experimental setup for each method as we use to generate the final results. We record the time taken to complete the first 30 steps of training, and use that to estimate the throughput (number of samples processed per second) and the total training time based on the total number of steps used to generate the final results listed in Table 4 and Table 12 (2000 steps for DP-GRAPE, 3000 steps for DP-Adam, and 20,000 steps for DPZero, as listed in Zhang et al. (2024).

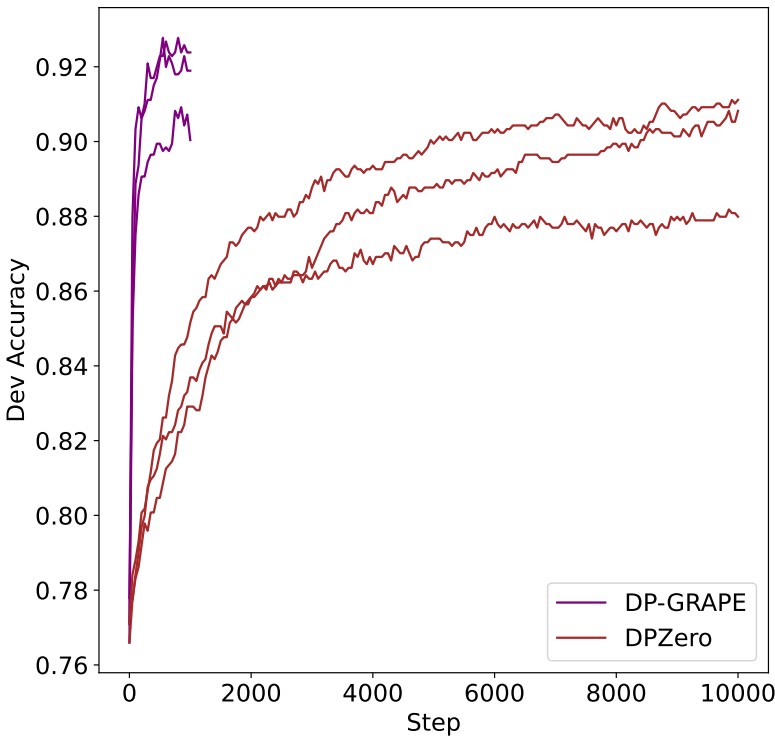

Figure 5: Convergence (as measured by development set accuracy) when fine-tuning RoBERTa-Large on SST-2 for DP-GRAPE and DPZero, with runs for three different random seeds used to generate few-shot datasets shown.

Table 11: Clipping parameter $C$ selected for each 1.3B/2.7B/6.7B OPT models and different datasets, for DP-Adam and DP-GRAPE. DP-Adam runs out of memory for the 6.7B models on all datasets.

|  | SST-2 | BoolQ | SQuAD | DROP |
|---|---|---|---|---|
| DP-Adam | 20.0/20.0/− | 5.0/20.0/− | 5.0/5.0/− | 5.0/20.0/− |
| DP-GRAPE | 20.0/20.0/1.0 | 20.0/20.0/5.0 | 0.1/0.1/0.1 | 0.1/0.1/1.0 |

Table 12: Mean and standard error of final f1 score over three different seeds for few-shot ($k = 1000$) fine-tuning of OPT models on SQuAD and DROP generation tasks, for different DP and non-private methods. The best DP result for each privacy level and dataset is in bold. OOM indicates out of memory on an 80GB GPU with a batch size of 1 and gradient accumulation.

| Model | OPT-1.3B | | OPT-2.7B | | OPT-6.7B | |
|---|---|---|---|---|---|---|
| Task | SQuAD | DROP | SQuAD | DROP | SQuAD | DROP |
| MeZO (non-private) | $73.5 \pm 1.2$ | $24.4 \pm 0.2$ | $76.3 \pm 0.8$ | $25.5 \pm 1.2$ | $79.7 \pm 1.1$ | $28.8 \pm 0.7$ |
| DPZero ($\varepsilon = 6$) | $72.6 \pm 0.8$ | $24.7 \pm 1.0$ | $75.7 \pm 1.5$ | $24.6 \pm 0.5$ | $\mathbf{79.5 \pm 0.9}$ | $\mathbf{28.4 \pm 1.3}$ |
| DPZero ($\varepsilon = 2$) | $70.1 \pm 1.6$ | $23.9 \pm 1.2$ | $71.9 \pm 1.2$ | $23.1 \pm 0.9$ | $77.1 \pm 1.0$ | $27.6 \pm 0.7$ |
| DP-Adam ($\varepsilon = 6$) | $76.9 \pm 0.2$ | $25.9 \pm 1.2$ | $81.4 \pm 0.7$ | $\mathbf{26.3 \pm 1.1}$ | OOM | OOM |
| DP-Adam ($\varepsilon = 2$) | $74.1 \pm 0.2$ | $\mathbf{25.2 \pm 1.8}$ | $77.8 \pm 0.4$ | $\mathbf{24.9 \pm 0.7}$ | OOM | OOM |
| DP-GRAPE ($\varepsilon = 6$) | $\mathbf{77.2 \pm 0.1}$ | $\mathbf{26.1 \pm 1.3}$ | $\mathbf{82.0 \pm 0.3}$ | $25.5 \pm 1.2$ | $79.5 \pm 0.2$ | $28.2 \pm 1.3$ |
| DP-GRAPE ($\varepsilon = 2$) | $\mathbf{76.7 \pm 0.7}$ | $25.0 \pm 1.1$ | $\mathbf{79.2 \pm 0.7}$ | $24.1 \pm 0.3$ | $\mathbf{77.6 \pm 0.4}$ | $\mathbf{27.8 \pm 0.6}$ |
| Zero-Shot | 26.8 | 11.1 | 29.8 | 9.7 | 36.5 | 17.8 |

Table 13: Throughput and total training time for fine-tuning on SQuAD with OPT models with a total batch size of 8 on an H100 GPU. Total training time is based on 2000 total steps for DP-GRAPE, 3000 total steps for DP-Adam (both are the same total number of steps we use to generate the results in Table 12) and 20000 total steps for DPZero, (the number of steps reported to generate the final results in Zhang et al. (2024). OOM indicates out of memory on an 80GB GPU with a batch size of 1 and gradient accumulation.

| Model | OPT-1.3B | | OPT-2.7B | | OPT-6.7B | |
|---|---|---|---|---|---|---|
| | Throughput (Samples/s) | Total Train Time (hours) | Throughput (Samples/s) | Total Train Time (hours) | Throughput (Samples/s) | Total Train Time (hours) |
| DPZero | 14.8 | 3.0 | 8.6 | 5.2 | 4.1 | 11.0 |
| DP-Adam | 9.6 | 0.7 | 4.4 | 1.5 | OOM | OOM |
| DP-GRAPE | 10.7 | 0.4 | 5.6 | 0.8 | 2.5 | 1.8 |

### C.4 HYPERPARAMETER RECOMMENDATIONS

Although we performed somewhat extensive hyperparameter searches for our different experiments, in many cases, large searches may be computationally infeasible, so here we give general recommendations for selecting a learning rate, subspace dimension $r$, subspace change frequency $F$, and clipping parameter $C$ when training with DP-GRAPE. For the learning rate, we found that a good range for fine-tuning with DP-GRAPE was between 1e-4 and 1e-3 (slightly larger than DP-Adam). In general, based on our experiments and previous works which use gradient projection, practitioners should use a larger subspace dimension $r$ for pre-training than for fine-tuning, and they should use a larger $r$ for larger models. For example, for non-private GaLore Zhao et al. (2024), $r = 128$ was used for pre-training a 60M Llama model, while $r = 512$ was used for pre-training a 1B Llama model, and $r = 4$ or $r = 8$ was used for fine-tuning a RoBERTa-Base model [1]. We used a similar range for $r$ with respect to the different models sizes we applied DP-GRAPE to. A subspace change frequency of $F = 100$ worked well for all of the fine-tuning experiments. We recommend practitioners do a search over clipping values between roughly $C = 0.1$ and $C = 100$, as the best clipping parameter can vary for different tasks.

### C.5 COMPUTATIONAL RESOURCES

We run all experiments on a single H100 GPU (although up to 4 were used at any one time to run separate experiments in parallel).

## D PRIVACY AND CONVERGENCE ANALYSIS

In this section, we provide a formal statement of Theorem 4.1 discussed in the main paper and its proof. Before the formal statement, we would like to give a more general version of the DP-GRAPE algorithm discussed in the main paper. Instead of considering a partition of gradients in DP-GRAPE which denotes each as gradients corresponding to a layer, we consider any general partition of the gradient vector in which we would independently project each part. This serves as a generalization as one can also introduce partitioning that is not necessarily demarcated by layers but may be something different (like the first column vector across layers). The generalized version of DP-GRAPE is given by Algorithm 6.

---

**Algorithm 6** Generalized Version of DP-GRAPE

---

**Require:** Dataset $X = \{\xi_1, \ldots, \xi_n\}$, batch size $B$, number of blocks (layers) $L$ which is a partition $\mathcal{U}_1, \mathcal{U}_2, \cdots, \mathcal{U}_L$ such that $\mathcal{U}_1 \cup \mathcal{U}_2 \cup \cdots \cup \mathcal{U}_L = [d]$ $|\mathcal{U}_\ell| = m_\ell n_\ell$ (hence $d = \sum_{\ell=1}^{L} m_\ell n_\ell$), initialization $w_0 \in \mathbb{R}^d$, number of iterations $T$, stepsize $\eta > 0$, clipping threshold $C > 0$, privacy parameters $\varepsilon > 0, \delta \in (0, 1)$.

1: Compute privacy noise variance $\sigma = \frac{2C\sqrt{T \log(1/\delta)}}{n\epsilon}$
2: **for** $t = 0, 1, \cdots, T - 1$ **do**
3:     Sample B data-points $X^t = \{\xi_j^t\}_{j=1}^{B}$ uniformly from $X$.
4:     **for** $\ell = 1, \cdots, L$ **do**
5:         For $i \in [r]$, sample $p_{\ell,i}^t$ i.i.d from $\mathcal{N}\left(0, \frac{1}{r}I_{m_\ell}\right)$ and define $P_\ell^t = [p_{\ell,1}^t \cdots p_{\ell,r}^t] \in \mathbb{R}^{|\mathcal{U}_\ell| \times r}$.
6:         For $j \in [B]$, compute projected gradient $R_{\ell,j}^t \leftarrow P_\ell^{t\top}(\nabla f(w_t; \xi_j^t)[\mathcal{U}_\ell])$.
7:     **end for**
8:     **for** $\ell \in [L]$ **do**
9:         Define $R_j^t = [R_{1,j}^t, \cdots, R_{L,j}^t] \in \mathbb{R}^{r \times L}$
10:        Privatize projected gradient $\tilde{R}^t[\mathcal{U}_\ell] = \frac{1}{B}\sum_{j=1}^{B} \text{clip}(R_j^t, C)[\mathcal{U}_\ell] + z_t$, where $z_t \sim \mathcal{N}(0, \sigma^2 \mathbf{I}_r)$.
11:        Update parameters: $w_{t+1}[\mathcal{U}_\ell] \leftarrow w_t[\mathcal{U}_\ell] - \eta P_\ell^t \tilde{R}^t[\mathcal{U}_\ell]$
12:    **end for**
13: **end for**
14: **Return** $w_\tau$ for $\tau$ sampled uniformly at random from $\{0, 1, \cdots, T - 1\}$.

---

We now give a brief overview of how Algorithm 6 works. We start with a fixed partition of the gradient vector, calling each partition a block. In DP-GRAPE, each block corresponds to a layer.

At each step, Algorithm 6 samples a batch of $B$ data points and computes a block-wise random projection of gradients using $r$ Gaussian vectors. It then aggregates gradients across blocks, clips them, and adds Gaussian noise to ensure DP. Finally, the previously sampled $r$ vectors are used to map updates back to the $d$-dimensional space, and parameters are updated. This is repeated for $T$ rounds.

Algorithm 6 is a more generalized version of DP-GRAPE, where each $\mathcal{U}_\ell$ for $\ell \in [L]$ represents indices in the flattened gradient vector corresponding to layer $\ell$. Thus, $|\mathcal{U}_\ell| = m_\ell n_\ell$. By replacing blocks with layer gradients, we see that projecting each block corresponds to projecting each layer's gradient. However, Algorithm 6 differs slightly from DP-GRAPE by using $r$ different projections of the flattened layer-wise gradients instead of left/right projections of gradient matrices. The method in DP-GRAPE has lower variance, leading to a similar upper bound on convergence.

To facilitate our analysis, we make the following assumptions:

**Assumption D.1** (Per-Sample Lipschitzness). The loss $f(\cdot; \xi)$ is $\Gamma$-Lipschitz for all $\xi \in X$, i.e. for all $w_1, w_2 \in \mathbb{R}^d$ $\|f(w_1; \xi) - f(w_2; \xi)\| \leq \Gamma \|w_1 - w_2\|$.

**Assumption D.2** (Smoothness). The average loss $F(w) := \frac{1}{n} \sum_{i=1}^{n} f(w; \xi_i)$ is $\lambda$-smooth for every given dataset $X$ i.e. for all $w_1, w_2 \in \mathbb{R}^d$, $\|\nabla F(w_1) - \nabla F(w_2)\| \leq \lambda \|w_1 - w_2\|$.

**Assumption D.3** (Finiteness of Optimal Value). $F^* := \min_{w \in \mathbb{R}^d} F(w)$ is finite.

*Remark 3.* It is important to note that these assumptions are standard in the analysis of private non-convex optimization (Lowy et al., 2024; Zhang et al., 2024).

Before presenting the complete proof of our theorem, we would like to define the following notations for the ease of stating our proofs.

**Notations and Lemmas:** For a set $A \subseteq X$, we define $f(w; A) = \frac{1}{|A|} \sum_{\xi \in A} f(w; \xi)$. By this definition, $F(w) = f(w; X)$. Moreover, for a vector $a \in \mathbb{R}^d$ and an ordered set $P = \{p_1, p_2, \cdots, p_q\} \subseteq [d]$, we define $a[P] := (a[p_1], a[p_2], \cdots a[p_q]) \in \mathbb{R}^q$ where $a[i]$ represents the $i^{th}$ index of the vector $a$. We assume that indexing starts from 1. $\|\cdot\|$ represents the $\ell_2$ norm while $\|\cdot\|_F$ represents the Frobenius norm. We now state some lemmas which would be useful throughout the proof.

**Lemma D.4.** *Consider any random variable $X \geq 0$ and an event $Q$, then we have that*

$$\mathbb{E}[X|Q] \leq \frac{\mathbb{E}[X]}{\mathbb{P}(Q)}$$

*Proof.* This directly follows from the law of total probability and the non-negativity of $X$

$$\mathbb{E}[X] = \mathbb{E}[X|Q]\mathbb{P}(Q) + \mathbb{E}[X|Q^c]\mathbb{P}(Q^c) \geq \mathbb{E}[X|Q]\mathbb{P}(Q)$$

which proves the given claim. $\square$

**Lemma D.5** (Greene (2003)). *Consider $X \sim \mathcal{N}(0, \sigma^2)$ then we have that*

$$\mathbb{E}[X^2|X \geq K] = \sigma^2 \left[1 + \frac{1}{\mathbb{P}[X \geq K]\sqrt{2\pi}} e^{-\frac{K^2}{2\sigma^2}}\right]$$

**Lemma D.6.** *Consider any random variable $X$ and a set $Q$ such that $\{X : X \geq t\} \subset Q$, then we get that*

$$\mathbb{E}[X|Q] \leq \mathbb{E}[X|X \geq t]$$

*Proof.* Using law of total probability, we get

$$\mathbb{E}[X|Q] = \mathbb{E}[X|Q, X \geq t]\mathbb{P}(X \geq t|Q) + \mathbb{E}[X|Q, X < t]\mathbb{P}(X < t|Q).$$

Since, $\{X : X \geq t\} \subset Q$, we have that $\mathbb{E}[X|Q, X \geq t] = \mathbb{E}[X|X \geq t]$. By conditioning, we have that $E[X|Q, X < t] < t \leq E[X|X \geq t]$ proving the given claim. $\square$

**Lemma D.7** (Lin et al. (2020)). *Let $\{a_l\}_{l \in [n]}$ be an arbitrary collection of vectors such that $\sum_{l=1}^{n} a_l = 0$. Further, let $\mathcal{S}$ be a uniformly random subset of $[n]$ of size $m$. Then,*

$$\mathbb{E}\left\|\frac{1}{m}\sum_{l \in \mathcal{S}} a_l\right\|^2 = \frac{n-m}{(n-1)m}\frac{1}{n}\sum_{l=1}^{n}\|a_l\|^2 \leq \frac{\mathbb{1}_{\{m<n\}}}{m\,n}\sum_{l=1}^{n}\|a_l\|^2.$$

**Lemma D.8** (Zhang et al. (2024))**.** *Let $u, v$ be uniformly sampled from the standard $d$-dimensional Gaussian, let $a \in \mathbb{R}^d$ be some fixed vector independent of $u$, and $H \in \mathbb{R}^{d \times d}$ be some fixed matrix independent of $u$. We have that*

(i) $\mathbb{E}[u] = 0$ *and* $\mathbb{E}[uu^\top] = \mathbf{I}_d$.

(ii) $\mathbb{E}_u[(u^\top a)u] = a$ *and* $\mathbb{E}_u[(u^\top a)^2 \|u\|^2] = d \|a\|^2$.

**Lemma D.9** (Theorem 1 of Abadi et al. (2016))**.** *There exist constants $c_1$ and $c_2$ so that given the number of steps $T$, batch size $B$, and sensitivity $\Delta$, for any $\varepsilon < c_1 \frac{B^2}{n^2}T$, the Gaussian Mechanism with noise level $\sigma$ applied for $T$ steps is $(\epsilon, \delta)$-differentially private for any $\delta > 0$ if we choose*

$$\sigma \geq c_2 \frac{B\Delta\sqrt{T \log(1/\delta)}}{n\epsilon}.$$

**Privacy-Utility Tradeoff:** Now, we state the main convergence result:

**Theorem D.10** (Formal version of Theorem 4.1)**.** *For any $\varepsilon > 0$ and $\delta \in (0, 1)$, Algorithm 6 is $(\varepsilon, \delta)$-DP. Under Assumptions D.1, D.2, D.3 and the fact that $\max_{0 \leq t \leq T} |F(w_t) - F^*| \leq D$, there exist a set of parameters such that the output $w_\tau$ satisfies*

$$\mathbb{E} \|\nabla F(w_\tau)\|^2 \leq \left( \lambda\sqrt{L}D + \left( 2 + 2\sqrt{L}\log\left( \left( 5\Gamma^2 + \frac{64dD\lambda}{r} \right) \frac{2\sqrt{2Ld}n^2\varepsilon}{\sqrt{\log(1/\delta)}} \right) \right) \Gamma^2 \right) \frac{4\sqrt{2d\log(1/\delta)}}{n\varepsilon}.$$

*Proof.* **Privacy Guarantee.** The proof of privacy follows by the fact that the sensitivity when one data point is replaced is given by $\Delta = \frac{2C}{B}$. Then we use Lemma D.9 with a sampling factor of $\frac{B}{n}$ and explicitly get the constants from Bassily et al. (2019) to get the value of $\sigma = \frac{2C\sqrt{T \log(1/\delta)}}{n\epsilon}$ in the algorithm for $\varepsilon \leq \frac{2B^2T}{n^2}$.

*Remark 4.* Note that in the above definition, we use the replace-one notion of sensitivity. But, the sensitivity for add-one/remove-one notion of DP still remains the same as the that we have mentioned above. For completion, we provide a short proof. Consider the per-sample vectors to be $v_1, \cdots, v_B$. Then, the sensitivity of the add-one/remove-one notion of DP would be $\Delta = \left| \frac{1}{n+1} \sum_{i=1}^{n+1} v_i - \frac{1}{n} \sum_{i=1}^{n} v_i \right| = \left| \frac{v_{n+1}}{n+1} - \frac{1}{n(n+1)} \sum_{j=1}^{n} v_j \right| \leq \frac{2C}{n}$. Thus, we get that the sensitivity upper bound for the replace-one and add/remove one notion of DP is exactly the same.

*Remark 5.* Note that we are essentially using a sub-sampled Gaussian mechanism in our algorithm which also satisfies other notions of DP such as Truncated Concentrated DP Bun et al. (2018) and Gaussian DP Dong et al. (2022). Hence, we can convert the bounds in terms of the parameters for different notions of privacy by getting the variance of the Gaussian noise in those parameters.

**Utility guarantee.** We focus on the utility guarantee on $\mathbb{E} \|\nabla F(w_\tau)\|^2$. Before going into the details of the proof, we would like to define some notations which would make things easier for us in the proof. For $A \subseteq X$ Define $\nabla f(w; A)[U_\ell] = \nabla_\ell f(w; A)$ and w.l.o.g. take $\nabla f(w, A) = (\nabla_1 f(w; A)^T, \nabla_2 f(w; A)^T, \cdots, \nabla_L f(w; A)^T)^T \in \mathbb{R}^d$. The same can be extended for $F(w) = f(w; X)$.

For any $t \in \{0, \cdots, T-1\}, i \in [r], j \in [n]$, and $\ell \in [L]$ consider the term $\left( \left( p_{\ell,i}^t \right)^\top \nabla_\ell f(w_t; \xi_j) \right)^2$. Since $p_{\ell,i}^t \sim \mathcal{N}\left( 0, \frac{1}{r}I_{m_\ell} \right)$. By Lipschitzness of $f(w_t; \xi_j)$ and the fact that $U_\ell \subseteq [d]$, we have that $\left( \left( p_{\ell,i}^t \right)^\top \nabla_\ell f(w_t; \xi_j) \right)^2 = \frac{\|\nabla_\ell f(w_t; \xi_j)\|^2}{r} \left( V_{\ell,i,j}^t \right)^2 \leq \frac{\Gamma^2}{r} \left( V_{\ell,i,j}^t \right)^2$ where $V_{\ell,i,j}^t \sim \mathcal{N}(0,1)$. Let $Q_{\ell,i,j}^t$ be the event such that $\left| V_{\ell,i,j}^t \right| \leq \frac{C}{\sqrt{L}\Gamma}$. Let $Q_t = \bigcap_{i=1}^{r} \bigcap_{j=1}^{n} \bigcap_{\ell=1}^{L} Q_{\ell,i,j}^t$. Thus, the probability that clipping does not happen at one iteration is greater than $\mathbb{P}(Q_t)$. We also denote $Q = \bigcap_{t=0}^{T-1} Q_t$. Hence, $\bar{Q} = \bigcup_{t=0}^{T-1} \bigcup_{i=1}^{r} \bigcup_{j=1}^{n} \bigcup_{\ell=1}^{L} \bar{Q}_{\ell,i,j}^t$ and for some $V \sim \mathcal{N}(0,1)$ we have that

$$\mathbb{P}\left[\left|V_{\ell,i,j}^t\right| \ge \frac{C}{\sqrt{L}\Gamma}\right] = \mathbb{P}\left[|V| \ge \frac{C}{\sqrt{L}\Gamma}\right] \text{ for some } V \sim \mathcal{N}(0,1). \text{ By the union bound, we have that}$$

$$\mathbb{P}(\bar{Q}) \le TLnr\mathbb{P}\left[|V| \ge \frac{C}{\sqrt{L}\Gamma}\right].$$

To simplify the notation, we let $G(x_t)$ represent

$$G_\ell(w_t) = \frac{1}{n}\sum_{j=1}^n\sum_{i=1}^r p_{\ell,i}^t \left(p_{\ell,i}^t\right)^\top \nabla_\ell f(w_t;\xi_j) = \sum_{i=1}^r p_{\ell,i}^t \left(p_{\ell,i}^t\right)^\top \nabla_\ell F(w_t),$$

For all $\ell \in [L]$, let $\hat{G}_\ell(w_t;X_t) = \frac{1}{B}P_\ell^t\left(\sum_{j=1}^B \text{clip}(R_j^t,C)[\ell]\right)$ and let

$$G_\ell(w_t;X_t) = \frac{1}{B}\sum_{j=1}^B\sum_{i=1}^r p_{\ell,i}^t \left(p_{\ell,i}^t\right)^\top \nabla_\ell f(w_t;x_j^t).$$

Note that the definition of $G_\ell$ is agnostic to whether clipping happens of not. But, conditioned on the event that clipping does not happen ($Q$), we have that $\hat{G}_\ell(w_t;X_t) = G_\ell(w_t;X_t)$.

Algorithm 6 becomes $X_t \sim Unif(X)$, $w_{t+1}[U_\ell] = w_t[U_\ell] - \eta(\hat{G}_\ell(w_t;X_t) + P_\ell^t z_t)$ under the above notation. Let $\hat{G}(w_t;X_t) = (\hat{G}_1(w_t;X_t)^\top,\cdots,\hat{G}_L(w_t;X_t)^\top)^\top$ and $P_t = \left[(P_1^t)^\top,\cdots,(P_L^t)^\top\right]^\top \in \mathbb{R}^{d\times r}$. Then, by smoothness of $F(w)$, we get that

$$F(w_{t+1}) \le F(w_t) - \eta\nabla F(w_t)^\top(\hat{G}(w_t;X_t) + P_t z_t) + \frac{\eta^2\lambda}{2}\left\|\hat{G}(w_t;X_t) + P_t z_t\right\|^2.$$

The event $Q_t$ depends on the randomness in $P_{<(t+1)} := \{P_0,\cdots,P_t\}$, $X_{<(t+1)} = \{X^0,\cdots,X^t\}$ and $z_{<t} := \{z_0,z_1,\cdots,z_{t-1}\}$. Note that the noise $z_t$ sampled from $\mathcal{N}(0,\sigma^2 I_r)$ is independent of $P_{<(t+1)}$, $X_{<(t+1)}$, $z_{<t}$, $w_t$, and the dataset $X$. Given that event $Q_t$ happens, it implies that we do not clip which implies that $\hat{G}(w_t;X_t) = G(w_t;X_t)$. Combining this with the fact that we clip per-sample, we get $\mathbb{E}_{X_{<(t+1)}}[\hat{G}(w_t;X_t)|Q_t] = \mathbb{E}_{X_{<(t+1)}}[G(w_t;X_t)|Q_t] = \mathbb{E}_{X_{<t}}[G(w_t)|Q_t]$. Conditioned on the event $Q_t$ and taking expectation with respect to $z_{<(t+1)}$, $X_{<(t+1)}$ and $P_{<(t+1)}$, we have that

$$\mathbb{E}_{z_{<(t+1)},X_{<(t+1)},P_{<(t+1)}}[F(w_{t+1})|Q_t] \le \mathbb{E}_{z_{<(t+1)},X_{<(t+1)},P_{<(t+1)}}[F(w_t)|Q_t]-$$

$$\eta\mathbb{E}_{z_{<(t+1)},X_{<(t+1)},P_{<(t+1)}}\left[\nabla F(w_t)^\top(\hat{G}(w_t;X_t) + P_t z_t)\Big|Q_t\right]$$

$$+ \frac{\eta^2\lambda}{2}\mathbb{E}_{z_{<(t+1)},X_{<(t+1)},P_{<(t+1)}}\left[\left\|\hat{G}(w_t;X_t) + P_t z_t\right\|^2\Big|Q_t\right]$$

$$= \mathbb{E}_{z_{<t},X_{<t},P_{<t}}[F_S(x_t)|Q_t]$$

$$- \eta\underbrace{\mathbb{E}_{z_{<t},X_{<t}P_{<(t+1)}}\left[\nabla F(w_t)^\top G(w_t)\big|Q_t\right]}_{①}$$

$$+ \frac{\eta^2\lambda}{2}\underbrace{\mathbb{E}_{z_{<t},X_{<(t+1)},P_{<(t+1)}}\left[\|G(w_t;X_t)\|^2\Big|Q_t\right]}_{②}$$

$$+ \frac{\eta^2\lambda}{2}\underbrace{\mathbb{E}_{z_{<(t+1)},X_{<t},P_{<(t+1)}}\left[z_t^\top P_t^\top P_t z_t\big|Q_t\right]}_{③}.$$

$$(3)$$

For term $①$, we get that

$$\mathbb{E}_{z_{<t},X_{<t},P_{<(t+1)}}\left[\nabla F(w_t)^\top\hat{G}(w_t)\Big|Q_t\right] = \sum_{\ell=1}^L \mathbb{E}_{z_{<t},X_{<t},P_{<(t+1)}}\left[\nabla_\ell F(w_t)^\top G_\ell(w_t)\,\big|\,Q_t\right].$$

By the law of total probability and Lemma D.8, since $P_t$ is independent of $w_t$, for each $\ell \in [L]$ we know that

$$\mathbb{E}_{z_{<t}, X_{<t}, P_{<(t+1)}} \left[ \nabla_\ell F(w_t)^\top G_\ell(w_t) \,\middle|\, Q_t \right] \mathbb{P}(Q_t) + \mathbb{E}_{z_{<t}, X_{<t}, P_{<(t+1)}} \left[ \nabla_\ell F(w_t)^\top G_\ell(w_t) \,\middle|\, \bar{Q}_t \right] \mathbb{P}(\bar{Q}_t)$$

$$= \mathbb{E}_{z_{<t}, X_{<t}, P_{<(t+1)}} \left[ \nabla_\ell F(w_t)^\top G_\ell(w_t) \right] = \mathbb{E}_{z_{<t}, X_{<t}, P_{<t}} \left[ \|\nabla_\ell F(w_t)\|^2 \right],$$

Rearranging terms, we thus obtain that

$$\mathbb{E}_{z_{<t}, X_{<t}, P_{<(t+1)}} \left[ \nabla_\ell F(w_t)^\top G_\ell(w_t) \,\middle|\, Q_t \right] = \frac{\mathbb{E}_{z_{<t}, X_{<t}, P_{<t}} \|\nabla_\ell F(w_t)\|^2}{\mathbb{P}(Q_t)} - \frac{\mathbb{E}_{z_{<t}, X_{<t}, P_{<(t+1)}} \left[ \nabla_\ell F(w_t)^\top G_\ell(w_t) \,\middle|\, \bar{Q}_t \right] \mathbb{P}(\bar{Q}_t)}{\mathbb{P}(Q_t)} \tag{4}$$

.

Using the definition of our event $\bar{Q}_t = \bigcup_{i=1}^r \bigcup_{j=1}^n \bigcup_{\ell=1}^L \bar{Q}^t_{\ell,i,j}$, we have that $\bar{Q}^t_{\ell,i,j} \subset \bar{Q}_t$. Thus, we have that,

$$\mathbb{E}_{z_{<t}, X_{<t}, P_{<(t+1)}} \left[ \nabla_\ell F(w_t)^\top G_\ell(w_t) \middle| \bar{Q}_t \right] = \sum_{i=1}^r \mathbb{E}_{z_{<t}, X_{<t}, P_{<(t+1)}} \left[ \left( (p^t_{\ell,i})^\top \nabla_\ell F(w_t) \right)^2 \middle| \bar{Q}_t \right]$$

$$\leq \frac{1}{n} \sum_{j=1}^n \sum_{i=1}^r \mathbb{E}_{z_{<t}, X_{<t}, P_{<(t+1)}} \left[ \left( (p^t_{\ell,i})^\top \nabla_\ell f(w_t; \xi_j) \right)^2 \middle| \bar{Q}_t \right]$$

$$= \frac{1}{nr} \sum_{j=1}^n \sum_{i=1}^r \|\nabla_\ell f(w_t; \xi_j)\|^2 \, \mathbb{E}_{z_{<t}, X_{<t}, P_{<(t+1)}} \left[ \left( V^t_{\ell,i,j} \right)^2 \middle| \bar{Q}_t \right]$$

$$\leq \frac{1}{rn} \sum_{j=1}^n \sum_{i=1}^r \|\nabla_\ell f(w_t; \xi_j)\|^2 \, \mathbb{E} \left[ \left( V^t_{\ell,i,j} \right)^2 \middle| \bar{Q}^t_{\ell,i,j} \right]$$

$$= \frac{1}{rn} \sum_{j=1}^n \sum_{i=1}^r \|\nabla_\ell f(w_t; \xi_j)\|^2 \, \mathbb{E} \left[ \left( V^t_{\ell,i,j} \right)^2 \middle| \left| V^t_{\ell,i,j} \right| \geq \frac{C}{\sqrt{L}\Gamma} \right]$$

$$= \frac{1}{n} \sum_{j=1}^n \|\nabla_\ell f(w_t; \xi_j)\|^2 \left[ 1 + \frac{1}{\mathbb{P} \left[ |V| \geq \frac{C}{\sqrt{L}\Gamma} \right] \sqrt{2\pi}} e^{-\frac{C^2}{2L\Gamma^2}} \right] . \tag{5}$$

The first equality uses the definition of $G_\ell$, the second inequality is Young's Inequality, the third equality uses the fact that $a^\top Z \sim \mathcal{N}(0, \|a\|^2)$ for $Z \sim \mathcal{N}(0, I_d)$ and $a \in \mathbb{R}^d$ independent of $Z$. The fourth and inequality uses Lemma D.6 with X as $\left( V^t_{\ell,i,j} \right)^2$. The fifth equality uses the fact that $\left( V^t_{\ell,i,j} \right)^2 \geq \frac{C^2}{L\Gamma^2}$ is equivalent to $\left| V^t_{\ell,i,j} \right| \geq \frac{C}{\sqrt{L}\Gamma}$. The sixth equality used Lemma D.5. Combining 5 with 4, we obtain that

$$E_{z_{<t}, X_{<t}, P_{<(t+1)}} \left[ \nabla_\ell F(w_t)^\top G_\ell(w_t) \,\middle|\, Q_t \right] \geq \frac{\mathbb{E}_{z_{<t}, X_{<t}, P_{<t}} \|\nabla_\ell F(w_t)\|^2}{2\,\mathbb{P}(Q_t)}$$

$$- \frac{1}{n} \sum_{j=1}^n \frac{\|\nabla_\ell f(w_t; \xi_j)\|^2 \, \mathbb{P}(\bar{Q}_t)}{\mathbb{P}(Q_t)} \left[ 1 + \frac{1}{\mathbb{P} \left[ |V| \geq \frac{C}{\sqrt{L}\Gamma} \right] \sqrt{2\pi}} e^{-\frac{C^2}{2L\Gamma^2}} \right] .$$

Thus, taking the sum over all $\ell \in [L]$, we get that

$$
\begin{aligned}
E_{z_{<t}, X_{<t}, P_{<(t+1)}} \left[ \nabla F(w_t)^\top G(w_t) \,\middle|\, Q_t \right] &\geq \sum_{\ell=1}^{L} \frac{\mathbb{E}_{z_{<t}, X_{<t}, P_{<t}} \|\nabla_\ell F(w_t)\|^2}{2\,\mathbb{P}(Q_t)} \\
&\quad - \frac{1}{n}\sum_{j=1}^{n}\sum_{\ell=1}^{L} \frac{\|\nabla_\ell f(w_t;\xi_j)\|^2 \, \mathbb{P}(\bar{Q}_t)}{\mathbb{P}(Q_t)} \left[ 1 + \frac{1}{\mathbb{P}\left[|V| \geq \frac{C}{\sqrt{L}\Gamma}\right]\sqrt{2\pi}} e^{-\frac{C^2}{2L\Gamma^2}} \right] \\
&= \frac{\mathbb{E}_{z_{<t}, X_{<t}, P_{<t}} \|\nabla F(w_t)\|^2}{2\,\mathbb{P}(Q_t)} \\
&\quad - \frac{1}{n}\sum_{j=1}^{n} \frac{\|\nabla f(w_t;\xi_j)\|^2 \, \mathbb{P}(\bar{Q}_t)}{\mathbb{P}(Q_t)} \left[ 1 + \frac{1}{\mathbb{P}\left[|V| \geq \frac{C}{\sqrt{L}\Gamma}\right]\sqrt{2\pi}} e^{-\frac{C^2}{2L\Gamma^2}} \right] \\
&\geq \frac{\mathbb{E}_{z_{<t}, X_{<t}, P_{<t}} \|\nabla F(w_t)\|^2}{2\,\mathbb{P}(Q_t)} - \frac{\Gamma^2 \mathbb{P}(\bar{Q}_t)}{\mathbb{P}(Q_t)} \left[ 1 + \frac{1}{\mathbb{P}\left[|V| \geq \frac{C}{\sqrt{L}\Gamma}\right]\sqrt{2\pi}} e^{-\frac{C^2}{2L\Gamma^2}} \right]
\end{aligned}
\tag{6}
$$

The second equality comes from the fact that the full gradient of the model is just concatenated version of the layer wise gradients. The third inequality follows from the per-sample Lipschitzness assumption D.1.

For term $②$, by the definition of $G(w_t; X_t)$ and using Lemma D.4, we have

$$
\mathbb{E}_{z_{<t}, X_{<(t+1)}, P_{<(t+1)}} \left[ \|G(w_t; X_t)\|^2 \middle| Q_t \right] \leq \sum_{\ell=1}^{L} \frac{\mathbb{E}_{z_{<t}, X_{<(t+1)}, P_{<(t+1)}} \left[ \|G_\ell(w_t; X_t)\|^2 \right]}{\mathbb{P}(Q_t)}
$$

$$
\begin{aligned}
\mathbb{E}_{z_{<t}, X_{<(t+1)}, P_{<(t+1)}} \left[ \|G_\ell(w_t; X_t)\|^2 \right] &\leq 2\,\mathbb{E}_{z_{<t}, X_{<t}, P_{<(t+1)}}[\|G_\ell(w_t)\|^2] + \\
&\quad 2\,\mathbb{E}_{z_{<t}, X_{<(t+1)}, P_{<(t+1)}}[\|G_\ell(w_t; X_t) - G_\ell(w_t)\|^2] \\
&= 2\mathbb{E}_{z_{<t}, X_{<t}, P_{<(t+1)}} \left[ \left\| \sum_{i=1}^{r} \left(p_{\ell,i}^t\right)^T \nabla_\ell F(w_t) p_{\ell,i}^t \right\|^2 \right] + \\
&\quad 2\mathbb{E}_{z_{<t}, X_{<(t+1)}, P_{<(t+1)}} \left[ \left\| \sum_{i=1}^{r} \left(p_{\ell,i}^t\right)^T \left(\nabla_\ell f(w_t; X_t) - \nabla_\ell F(w_t)\right) p_{\ell,i}^t \right\|^2 \right] \\
&= \left( \frac{d+r-1}{r} \right) \mathbb{E}_{z_{<t}, X_{<t}, P_{<t}}[\|\nabla_\ell F(w_t)\|^2] + \\
&\quad \left( \frac{d+r-1}{r} \right) \mathbb{E}_{z_{<t}, X_{<(t+1)}, P_{<t}}[\|\nabla_\ell f(w_t; X_t) - \nabla_\ell F(w_t)\|^2] \\
&\leq \frac{2d}{r}\mathbb{E}_{z_{<t}, X_{<t}, P_{<t}}[\|\nabla_\ell F(w_t)\|^2] + \frac{2d\,\|\nabla_\ell F(w_t)\|^2 \,\mathbb{1}_{\{B<n\}}}{rB}
\end{aligned}
$$

Thus, taking the sum and using the fact that $\sum_{\ell=1}^{L} \|\nabla_\ell F(w_t)\|^2 = \|\nabla F(w_t)\|^2 \leq \Gamma^2$, we get that

$$
\mathbb{E}_{z_{<t}, X_{<(t+1)}, P_{<(t+1)}} \left[ \|G(w_t; X_t)\|^2 \middle| Q_t \right] \leq \frac{2d}{r\mathbb{P}(Q_t)}\mathbb{E}_{z_{<t}, X_{<t}, P_{<t}}[\|\nabla F(w_t)\|^2] + \frac{8d\Gamma^2 \mathbb{1}_{\{B<n\}}}{rB\mathbb{P}(Q_t)}
\tag{7}
$$

For term ③, $P_t$ is essentially a $d \times r$ matrix of independent $\mathcal{N}\left(0, \frac{1}{r}\right)$ entries. Since the identity matrix is positive semi-definite, using Lemma D.4 and Lemma D.8 we get that

$$
\begin{aligned}
\mathbb{E}_{z_{<(t+1)}, X_{<t}, P_{<(t+1)}}\left[z_t^\top P_t^\top P_t z_t \big| Q_t\right] &\leq \frac{\mathbb{E}_{z_{<(t+1)}, X_{<t}, P_{<(t+1)}}\left[z_t^\top P_t^\top P_t z_t\right]}{\mathbb{P}(Q_t)} \\
&= \frac{\sigma^2 \mathbb{E}_{z_{<t}, X_{<t}, P_{<(t+1)}}\left[Tr(P_t^\top P_t)\right]}{\mathbb{P}(Q_t)} \\
&= \frac{\sigma^2 E_{z_{<t}, X_{<t}, P_{<(t+1)}}\left[\sum_{i=1}^r \|p_{ti}\|^2\right]}{\mathbb{P}(Q_t)} \\
&= \frac{\sigma^2 d}{\mathbb{P}(Q_t)}.
\end{aligned}
\tag{8}
$$

Plugging (6), (7) and (8) back into (3), we obtain that

$$
\begin{aligned}
\mathbb{E}_{z_{<(t+1)}, X_{<(t+1)}, P_{<(t+1)}}[F(w_{t+1})|Q_t] \leq{}& \mathbb{E}_{z_{<t}, X_{<t}, P_{<t}}[F(w_t)|Q_t] \\
&- \frac{\eta}{2}\left(1 - \frac{2d\lambda\eta}{r}\right) \frac{\mathbb{E}_{z_{<t}, \xi_{<t}, U_{<t}}\|\nabla F(w_t)\|^2}{\mathbb{P}(Q_t)} \\
&+ \frac{\eta\Gamma^2 \mathbb{P}(\bar{Q}_t)}{\mathbb{P}(Q_t)}\left[1 + \frac{1}{\mathbb{P}\left[|V| \geq \frac{C}{\sqrt{L}\Gamma}\right]\sqrt{2\pi}}e^{-\frac{C^2}{2L\Gamma^2}}\right] \\
&+ \frac{4d\eta^2\Gamma^2\lambda \mathbb{1}_{m<n}}{Br\mathbb{P}(Q_t)} + \frac{\eta^2\lambda\sigma^2 d}{2\mathbb{P}(Q_t)}
\end{aligned}
$$

Assuming $\frac{2d\lambda\eta}{r} \leq 1$ and choosing $\eta \leq \frac{r}{4d\lambda}$, we have that

$$
\begin{aligned}
\mathbb{E}_{z_{<(t+1)}, X_{<(t+1)}, P_{<(t+1)}}\|\nabla F(w_t)\|^2 \leq{}& \frac{4\mathbb{E}_{z_{<(t+1)}, X_{<(t+1)}, P_{<(t+1)}}[F(w_t) - F(w_{t+1})|Q_t]\mathbb{P}(Q_t)}{\eta} \\
&+ 2\lambda\eta d\sigma^2 + \frac{16d\eta \mathbb{1}_{B<n}\Gamma^2\lambda}{Br} \\
&+ 4\eta\Gamma^2\left[1 + \frac{1}{\mathbb{P}\left[|V| \geq \frac{C}{\sqrt{L}\Gamma}\right]\sqrt{2\pi}}e^{-\frac{C^2}{2L\Gamma^2}}\right]\mathbb{P}(\bar{Q}).
\end{aligned}
$$

Recall $Q_t$ is the event that clipping does not happen at iteration $t$ and $Q$ is the event that clipping does not happen for every iteration, hence $Q_t \cap Q = Q$. By the law of total probability and the assumption that $|F(w_t) - F^*| \leq D$ for every $t$, we have that

$$
\begin{aligned}
\mathbb{E}_{z_{<(t+1)}, X_{<(t+1)} P_{<(t+1)}}[F(w_t) - F(w_{t+1})|Q_t]\mathbb{P}(Q_t) &= \mathbb{E}_{z_{<T}, X_{<T}, P_{<T}}[F(w_t) - F(w_{t+1})|Q_t]\mathbb{P}(Q_t) \\
&= \mathbb{E}_{z_{<T}, X_{<T}, P_{<T}}\left[F(w_t) - F(w_{t+1})\big| Q_t \cap Q\right]\mathbb{P}(Q_t \cap Q) \\
&\quad + \mathbb{E}_{z_{<T}, X_{<T}, P_{<T}}\left[F(w_t) - F(w_{t+1})\big| Q_t \cap \bar{Q}\right]\mathbb{P}(Q_t \cap \bar{Q}) \\
&\leq \mathbb{E}_{z_{<T}, X_{<T}, P_{<T}}[F(w_t) - F(w_{t+1})|Q]\mathbb{P}(Q) + 2D\,\mathbb{P}(\bar{Q}).
\end{aligned}
$$

As a result, we have that

$$
\begin{aligned}
\mathbb{E}_{z_{<(t+1)}, X_{<(t+1)}, P_{<(t+1)}}\|\nabla F(w_t)\|^2 \leq{}& \frac{4\mathbb{E}_{z_{<T}, X_{<T}, P_{<T}}[F(w_t) - F(w_{t+1})|Q]\mathbb{P}(Q)}{\eta} \\
&+ 2\lambda\eta d\sigma^2 + \frac{16\eta \mathbb{1}_{B<n}\Gamma^2\lambda}{Br} \\
&+ \left(4d\eta\Gamma^2\left[1 + \frac{1}{\mathbb{P}\left[|V| \geq \frac{C}{\sqrt{L}\Gamma}\right]\sqrt{2\pi}}e^{-\frac{C^2}{2L\Gamma^2}}\right] + \frac{8D}{\eta}\right)\mathbb{P}(\bar{Q}).
\end{aligned}
$$

Taking expectation with respect to all randomness, i.e., $\mathbb{E} = \mathbb{E}_{z_{<T}, \xi_{<T}, u_{<T}}$, summing up from $t = 0$ to $T - 1$, and dividing both sides by $T$, we have that

$$\mathbb{E} \left\| \nabla F(w_\tau) \right\|^2 = \frac{1}{T} \sum_{t=1}^{T} \mathbb{E}_{z_{<t}, \xi_{<t}, u_{<t}} \left\| \nabla F(w_t) \right\|^2$$

$$\leq \frac{4\,\mathbb{E}[F(w_0) - F(w_T)|Q]\mathbb{P}(Q)}{\eta T} + \frac{8\lambda\eta T\, d\log(1/\delta)}{n^2\varepsilon^2}C^2 + \frac{16 d\eta \mathbb{1}_{B<n}\Gamma^2\lambda}{Br}$$

$$+ \left( 4\eta\Gamma^2 \left[ 1 + \frac{1}{\mathbb{P}\left[ |V| \geq \frac{C}{\sqrt{L}\Gamma} \right]\sqrt{2\pi}} e^{-\frac{C^2}{2L\Gamma^2}} \right] + \frac{8D}{\eta} \right) \mathbb{P}(\bar{Q})$$

$$\leq \frac{4\,\mathbb{E}[F(w_0) - F(w_T)|Q]\mathbb{P}(Q)}{\eta T} + \frac{8\lambda\eta T\, d\log(1/\delta)}{n^2\varepsilon^2}C^2 + \frac{16 d\eta \mathbb{1}_{B<n}\Gamma^2\lambda}{Br}$$

$$+ \left( 4\eta\Gamma^2 \left[ \mathbb{P}\left[ |V| \geq \frac{C}{\sqrt{L}\Gamma} \right] + \frac{1}{\sqrt{2\pi}} e^{-\frac{C^2}{2L\Gamma^2}} \right] + \frac{8D}{\eta}\mathbb{P}\left[ |V| \geq \frac{C}{\sqrt{L}\Gamma} \right] \right) TLnr$$

$$\leq \frac{4\,\mathbb{E}[F(w_0) - F(w_T)|Q]\mathbb{P}(Q)}{\eta T} + \frac{8\lambda\eta T\, d\log(1/\delta)}{n^2\varepsilon^2}C^2$$

$$+ \frac{16 d\eta \mathbb{1}_{B<n}\Gamma^2\lambda}{Br} + \left( 4\left[ 2 + \frac{1}{\sqrt{2\pi}} \right]\eta\Gamma^2 + \frac{16D}{\eta} \right) TLnr e^{-\frac{C^2}{2L\Gamma^2}}$$

Considering the choice of parameters to be

$$\eta T = \frac{n\varepsilon}{\lambda\sqrt{2Ld\log(1/\delta)}}, \eta = \frac{r}{4d\lambda}, T = \frac{2\sqrt{2d}n\varepsilon}{r\sqrt{L\log(1/\delta)}},$$

$$C = \Gamma\sqrt{2L\log\left( \left( \frac{5r}{4d\lambda}\Gamma^2 + \frac{64dD\lambda}{r} \right)\frac{2\sqrt{2L}dn^2\varepsilon}{\sqrt{\log(1/\delta)}} \right)},$$

$$B \geq \max\left( \sqrt{\frac{r}{8n}}\left( \frac{L\log(1/\delta)}{2d} \right)^{1/4}, \frac{n\varepsilon}{2\sqrt{2d\log(1\delta)}} \right),$$

we get that,

$$\mathbb{E} \left\| \nabla F(w_\tau) \right\|^2 \leq \left( \lambda\sqrt{L}D + \left( 2 + 2\sqrt{L}\log\left( \left( 5\Gamma^2 + \frac{64dD\lambda}{r} \right)\frac{2\sqrt{2L}dn^2\varepsilon}{\sqrt{\log(1/\delta)}} \right) \right)\Gamma^2 \right)\frac{4\sqrt{2d\log(1/\delta)}}{n\varepsilon}.$$

$$\square$$

