# OpenReview forum: "Memory-Efficient Differentially Private Training with Gradient Random Projection"
_ICLR.cc/2026/Conference — Submitted to ICLR 2026_

### Official Review · Reviewer_gYPz · 2025-10-16

**Soundness:** 3
**Presentation:** 2
**Contribution:** 2
**Rating:** 4
**Confidence:** 3

**Summary:**

This paper introduces DP-GRAPE, a method to reduce the memory overhead of DP training by using random projections.

**Strengths:**

1. The method is supported by a theoretical analysis of its privacy and utility guarantees.

2. Empirical results show it effectively reduces memory usage while maintaining model accuracy.

**Weaknesses:**

The choice of the projection dimension r is a key hyperparameter, but the paper provides little guidance on how to set it. The trade-off between memory compression and model utility needs a systematic analysis.

**Questions:**

1. After projecting the gradient, how is the noisy gradient used to update the model parameters?

2. In the memory comparison (Table 2), why is DP-GRAPE's cost $n_l$ and not $m_ln_l$? Don't you need to compute the full gradient before projecting it?

3. The Theorem says DP-GRAPE achieve comparable trade-off to DP-SGD. Do you have any experimental results? Also, what is the memory usage of DP-SGD?

4. For the DP-Adam experiments, it is important to compare against the correct baseline, DP-AdamBC [1], which accounts for bias correction. Why was this comparison omitted?

[1] DP-AdamBC: Your DP-Adam Is Actually DP-SGD (Unless You Apply Bias Correction), AAAI'24

---

> ### Author Response · Authors · 2025-11-23
>
> We thank the reviewer for the time taken to read our paper and for the constructive and thoughtful feedback. Please find our detailed responses below.
>
>
> > W1: The choice of the projection dimension r is a key hyperparameter, but the paper provides little guidance on how to set it. The trade-off between memory compression and model utility needs a systematic analysis.
>
> We would like to highlight that we have provided the complete utility expression of our algorithm including the dependence of projection dimension $r$ on Page 27, Line 1437. This expression provides principled guidance for selecting the projection dimension $r$, indicating that beyond a certain point, further increases in $r$ offers negligible utility improvements (while significantly increasing memory usage).
>
> Empirically, in Figure 4 of Appendix C.2 in the revised upload, we provide additional plots showing RoBERTa fine-tuning accuracy on SST-2 for $r = 4, 8, 16, 32, 64, 128, 256$, using the same setup as in the experiments for Table 3. The results indicate that reducing $r$ below $32$ yields only marginal additional memory savings (as also shown in Figure 4), while the highest SST-2 accuracy is obtained at $r = 16$ for both tested privacy levels. Based on these observations, we recommend choosing $r \in [8, 32]$ for similar fine-tuning tasks.
>
> > Q1: After projecting the gradient, how is the noisy gradient used to update the model parameters?
>
> After projecting the gradient, the per-sample projected gradients are clipped, summed and then noise is added (lines 262-263, in Algorithm 1 block). The noisy gradient is then used to update the first- and second-order (projected) moment estimates for Adam (lines 277-278, in Algorithm 2 block). Finally, the model parameters are updated using the standard Adam update rule except with the additional step of projecting the update back to the full-dimensional space (line 281, in Algorithm 2 block).
>
> > Q2: In the memory comparison (Table 2), why is DP-GRAPE's cost $n_\ell$ and not $m_\ell n_\ell$? Don't you need to compute the full gradient before projecting it?
>
> For DP-GRAPE, the cost to store the projected gradient is $r n_\ell$, where r is the projection dimension. It is true that we have to compute the full gradient of size $m_\ell \times n_\ell$ before projecting it, however we implement a layerwise projection strategy where the gradient for each layer is projected immediately after it is computed during backpropagation, rather than after finishing the backwards pass (lines 198-200).
>
> > Q3: The Theorem says DP-GRAPE achieve comparable trade-off to DP-SGD. Do you have any experimental results? Also, what is the memory usage of DP-SGD?
>
> Thank you for raising this point. While we do not include experiments with DP-SGD, we would like to clarify why our technique does not yield significant memory savings in that setting. It is true that we gain some memory savings when computing the per-sample gradients in SGD—since these per-sample gradients are projected into a lower-dimensional subspace—but this has only a minor effect on the final memory footprint of the algorithm. Our method gains efficiency specifically because, under Adam, the intermediate momentum states can be maintained in a low-dimensional subspace and later reconstructed exactly in the full space. This relies on the MeZO-style seed-based reconstruction trick, where we never store the full projection matrix; instead, we store only the random seed that generates it. Since the projection matrix can be regenerated deterministically from the seed, we can work entirely in the low-dimensional space during training and recover the exact full-dimensional quantities when needed—producing true memory savings.
>
> DP-SGD, however, does not maintain momentum or multi-step states, so there is no analogous opportunity for compression and reconstruction, and the projection mechanism offers no significant memory benefit. For theoretical comparison, we benchmarked against standard differentially private SGD because under a constant learning rate, SGD is a special case of Adam with zero momentum, matching the structure assumed in our analysis. We appreciate the reviewer highlighting this issue and will make this reasoning clearer in the final version.

---

> ### Author Response · Authors · 2025-11-23
>
> > Q4: For the DP-Adam experiments, it is important to compare against the correct baseline, DP-AdamBC [1], which accounts for bias correction. Why was this comparison omitted?
>
> We appreciate the reviewer pointing out the DP-AdamBC baseline. We were not aware of this work earlier, and we thank the reviewer for bringing it to our attention. We will incorporate a comparison to DP-AdamBC in the revised version of our paper.
>
> However, we would also like to note that, based on the findings in the DP-AdamBC paper, a bias-corrected variant of our method can be developed in a very similar way. The main idea behind DP-AdamBC is to correct for the bias introduced by clipping and the additive DP noise in the first and second moment estimates. This bias appears because the noise affects the moving-average statistics, and the usual Adam denominator no longer properly normalizes these states. Since the privatization mechanism used in our method is the same as in DP-Adam (and therefore DP-AdamBC)—namely, clipping followed by Gaussian noise—the source of bias in the momentum terms is identical. As a result, the same correction strategy used in DP-AdamBC can be applied to DP-GRAPE: one can adjust the denominator to remove the expected contribution of the noise and recover an unbiased estimate of the second-moment statistics. Thus, implementing a bias-corrected version of DP-GRAPE is straightforward and would require only the same adjustment that DP-AdamBC applies.

---

> > ### Comment · Reviewer_gYPz · 2025-11-25
> > **Comment**
> >
> > Thank you for your response. My concern has been addressed, and I have updated the score to 6.

---

### Official Review · Reviewer_dFp7 · 2025-10-29

**Soundness:** 2
**Presentation:** 3
**Contribution:** 2
**Rating:** 4
**Confidence:** 4

**Summary:**

The paper proposes DP-GRAPE, a memory-efficient DP training method that projects each per-sample gradient into a low-dimensional random subspace, then performs clipping and Gaussian privatization after projection. This design cuts per-sample gradient and optimizer-state memory from full dimension to the projection dimension r, enabling large-model DP training without materializing full per-sample gradients. The authors argue SVD-based subspaces are unnecessary in the DP regime because privatization flattens the singular-value spectrums of gradients, so cheap Gaussian projections suffice and avoid expensive SVDs or storing projectors. Theoretically, under standard assumptions, the algorithm achieves (\epsilon,\delta)-DP and the expected stationarity gap matches that of DP-SGD up to log factors when the number of layers are considered as a constant. Empirically, it maintains accuracy comparable to first-order DP baselines while substantially reducing memory and scaling to larger models.

**Strengths:**

Originality. The paper advances DP training by coupling project-then-privatize gradient handling with random low-rank projections, motivated by the observation that privatization flattens the gradient spectrum.
Quality. The paper provides rigorous theoretical guarantees and offers reproducible implementation details and hyperparameter guidance.
Clarity. Figures, tables, and the presentation of the algorithm are clear with consistent notation that makes the method easy to follow.
Significance. DP-GRAPE substantially reduces the memory usage of DP training while preserving comparable accuracy.

**Weaknesses:**

Limited novelty (main concern).
Algorithmically, the core move—projecting gradients into a low-dimensional subspace and then privatizing—is a direct transplant of low-rank / random-projection ideas into the DP setting; the paper does not introduce a fundamentally new optimization principle. On the theory side, the guarantees largely read as an incremental generalization of standard DP-SGD analyses to the projected case.

Missing head-to-head experiments with the methods surveyed in Table 1.
Table 1 contrasts DP-SGD-JL[1], Ghost Clipping[2], and Book-Keeping[3], but the paper does not reproduce them under the same models/hardware/privacy accounting—leaving the table’s claims unsupported in this setting. In the zeroth-order line, only DPZero is included while DP-ZO[4] is omitted, which is a notable gap given the scarcity and relevance of Zeroth-order DP work.

Opaque memory attribution.
The comparisons do not decompose where memory is saved or spent—parameters, gradients, optimizer states, and activations—nor do they separate forward/backward/communication peaks. As a result, readers cannot tell whether the gains are dominated by optimizer-state shrinking, gradient tensor compression, or interactions with activation checkpointing.

[1]  Fast and memory efficient differentially private-sgd via jl projections. Advances in Neural Information Processing Systems, 34:19680–19691, 2021.
[2] Large language models can be strong differentially private learners. arXiv preprint arXiv:2110.05679, 2021.
[3] Differentially private optimization on large model at small cost. In International Conference on Machine Learning, pp. 3192–3218. PMLR, 2023.
[4] Private fine-tuning of large language models with zeroth-order optimization. arXiv preprint arXiv:2401.04343, 2024.

**Questions:**

Questions
Will you add DP-SGD-JL, Ghost Clipping, Book-Keeping, and DP-ZO under identical models and hardware, reporting peak memory, throughput, and wall-clock to a fixed validation target, so Table-1 claims are empirically supported?

Can you include a stacked memory breakdown separating parameters/gradients/ optimizer states/activations and phase-specific peaks (forward/backward/communication)?

Since the analysis only covers SGD-type convergence now, can you provide an Adam-style convergence guarantee of DP-GRAPE?

---

> ### Author Response · Authors · 2025-11-23
>
> We thank the reviewer for the time taken to read our paper and for the constructive and thoughtful feedback. Please find our detailed responses below.
>
> > W1: Limited novelty (main concern) ... case.
>
> We respectfully disagree with the claim that the novelty of our work is limited. While we do build on the idea of projecting gradients, our contribution is not a direct transplant of existing low-rank or random-projection techniques into the DP setting. To the best of our knowledge, no prior work has recognized or exploited the substantial memory savings that arise when privatization is applied after projecting gradients into a lower-dimensional space. This memory-efficiency perspective is central to our contribution.
> As described in the methodology section, our work is motivated by the observation that a direct DP adaptation of gradient-based low-rank methods such as GaLore does not work well in practice. In such methods, the subspace is derived from the gradients themselves via SVD. However, to maintain privacy, the gradients must be privatized before the SVD step. Privatization (i.e., clipping and adding noise) flattens the singular value spectrum, destroying the very low-rank structure the SVD attempts to capture. This makes SVD-based subspace learning ineffective under DP. Our approach instead uses random projections, which do not rely on gradient-derived structure and therefore remain stable after privatization. We experimentally show that, in the DP setting, gradient-based subspace methods do not scale well, whereas simple random projections can outperform them while providing significant memory savings.
>
> On the theoretical side, bounding the sensitivity of the projected gradient (see Theorem 4.1) is non-trivial. Allowing random projections introduces the possibility that the projected gradient can be unbounded in the worst case. Our analysis must therefore track the high-probability behavior of the projection and incorporate failure events carefully, ensuring that these do not cause meaningful utility degradation. This leads to a convergence analysis that, although structurally reminiscent of SGD, diverges substantially once gradient variance and projection-induced variability are accounted for.
> Even more importantly, our theoretical analysis differs significantly from existing work on DP optimization with clipping. The closest prior work is DPZero (Zhang et al., 2024), which provides the first memory-efficient DP optimization guarantee. In the smooth case, their finite-difference step corresponds to a rank-1 projection (r = 1) in our framework. Our analysis generalizes this to arbitrary $r \geq 1$. However, their method requires sampling projection vectors uniformly from the unit sphere, which necessitates normalizing high-dimensional Gaussian vectors—an operation known to be a computational bottleneck in DP optimization because computing norms incurs significant overhead (Bu et al., 2021). This issue arises not only for gradients but also for the projection vectors themselves, especially in block-wise implementations. Furthermore, Gaussian-based projections have unbounded worst-case norm, and clipping them increases the expected norm of the surviving samples. One workaround involves truncating each component of the Gaussian vector to lie within symmetric bounds (and adjusting for unbiasedness), which aligns with the framework of Zhang et al. (2024). However, this still leads to the same types of memory or runtime overheads as clipping because every vector component must be truncated individually.
>
> Our analysis overcomes these issues by addressing two main challenges: (1) managing the variance of layer-wise gradient projections, and (2) carefully controlling the expectations under failure events—i.e., ensuring that rare large deviations in projection norms do not harm convergence. Through careful probabilistic analysis, we show that although Gaussian projection vectors have unbounded worst-case norms, the tail probabilities decay exponentially fast. As a result, these rare events do not meaningfully affect convergence, even in expectation. This allows us to safely use unnormalized Gaussian vectors, bypassing the need for normalization or truncation, thereby avoiding the computational bottlenecks of the DPZero framework while still maintaining rigorous theoretical guarantees. We have discussed these points on Lines 296-340 (Pages 6 and 7).
>
> In summary, our contributions include not only a practically effective memory-efficient DP optimization algorithm but also a non-trivial theoretical framework that generalizes and improves upon prior analyses while addressing key computational challenges.

---

> ### Author Response · Authors · 2025-11-23
>
> > W2: Missing head-to-head experiments with the methods surveyed in Table 1. Table 1 contrasts DP-SGD-JL[1], Ghost Clipping[2], and Book-Keeping[3], but the paper does not reproduce them under the same models/hardware/privacy accounting—leaving the table’s claims unsupported in this setting. In the zeroth-order line, only DPZero is included while DP-ZO[4] is omitted, which is a notable gap given the scarcity and relevance of Zeroth-order DP work.
>
> > Q1: Will you add DP-SGD-JL, Ghost Clipping, Book-Keeping, and DP-ZO under identical models and hardware, reporting peak memory, throughput, and wall-clock to a fixed validation target, so Table-1 claims are empirically supported?
>
> We will add the numbers corresponding to Ghost Clipping and Bookkeeping for the models we have mentioned. It is worth noting that DP-ZO and DP-Zero are the same algorithm, a point made explicitly in both original papers. Consequently, any comparison we make with DP-Zero should be interpreted as a comparison with DP-ZO as well. For DP-SGD-JL, the gradient and its norm are estimated using random projections. In our setting, we can simply replace their gradients with our projected gradients and then apply their randomized clipping procedure. Conditioning on the projection matrix, we can directly reuse their privacy analysis to obtain the corresponding privacy guarantees for our method. Because our gradients live in an even lower-dimensional projected space, this approach would provide additional memory savings during norm estimation and clipping. We will add this comparison as well in the final version of our paper.
>
> Regarding implementation details, techniques such as bookkeeping and ghost clipping affect only how per-sample gradients are computed and privatized; they do not modify the optimization algorithm itself. Once gradients are computed using these techniques, they can be safely projected into a lower-dimensional space, after which any further privatization steps can be applied directly to the projected vectors. This is valid because our privacy analysis already conditions on knowledge of the projection matrix. As a result, incorporating bookkeeping or ghost clipping into DP-GRAPE does not alter the utility–privacy trade-off. In fact, ghost clipping can further reduce per-sample memory usage, with only a modest increase in per-iteration runtime. We have mentioned this in Page 9, lines 477 - 481.
>
> However, we would like to point out that to enable book-keeping and ghost clipping in our algorithm, we would need to clip using an upper bound on the projected gradient rather than the actual norm of the projected gradient. More concretely, if $M$ is a random matrix and $G$ is the per-sample gradient and $M^T G$ is the projected gradient update (we can concatenate all the layer-wise gradient matrices and corresponding random matrices), then we would need to clip by comparing $C$ with $|M|_F$ $|G|_F$  (where $|H|_F$ represents the Frobenius norm of a matrix)  instead of using the base projected gradient norm. This discrepancy can be corrected by increasing the learning rate by a factor that scales approximately with the square root of the total dimension random matrices divided by the per-entry standard deviation of the individual gaussian entries (at least in SGD) as we know that the value of ${|M|_F}^2$ would be tightly concentrated around its mean. Note that because we use the same projection matrix for every sample, we only need to compute the norm of the projection matrix once, and then we can compute the per-sample norms $|G|_F$ using techniques such as book-keeping and ghost clipping. Our convergence analysis does not change under this setting, as our high-probability event assumes that our upper bound is less than or equal to $C$ (and sets $C$ accordingly), rather than relying on concentration of the actual projected gradient norm. We will add a brief clarification on this in the final version of our paper.

---

> ### Author Response · Authors · 2025-11-23
>
> > W3: Opaque memory attribution. The comparisons do not decompose where memory is saved or spent—parameters, gradients, optimizer states, and activations—nor do they separate forward/backward/communication peaks. As a result, readers cannot tell whether the gains are dominated by optimizer-state shrinking, gradient tensor compression, or interactions with activation checkpointing.
>
> > Q2: Can you include a stacked memory breakdown separating parameters/gradients/ optimizer states/activations and phase-specific peaks (forward/backward/communication)?
>
> Thank you for the suggestion. We will include these numbers in the revised version of the paper. We would like to emphasize that the memory savings achieved by DP-GRAPE come from compressing per-sample gradients (and accumulated gradients, when gradient accumulation is used) and compressing optimizer states. Importantly, DP-GRAPE does not change the memory usage of model parameters or activations, so the improvements over DP-Adam arise solely from these two sources. We also clarify that we do not use activation checkpointing in any of our experiments.
>
> Whether the dominant memory savings come from optimizer-state shrinking or gradient compression depends on the model size. In optimizers such as Adam or AdamW, each parameter is associated with multiple auxiliary tensors (e.g., first and second moment estimates), often doubling or tripling the memory footprint relative to storing the parameters alone. As model size grows, these optimizer states scale proportionally and become one of the largest contributors to total memory consumption during training. In contrast, the relative size (compared to DP-Adam) of per-sample gradients for large models does not grow as aggressively. As a result, for large models, the memory savings obtained by compressing optimizer states become substantial—often dominating the overall reduction—because shrinking these high-dimensional moment tensors directly reduces a major memory bottleneck. For smaller models, the relative gains compared to DP-Adam will be dominated by the sample gradient compression. This is why effective optimizer-state compression, as done in DP-GRAPE, is especially impactful in the large-model regime.
>
> > Q3: Since the analysis only covers SGD-type convergence now, can you provide an Adam-style convergence guarantee of DP-GRAPE?
>
> Thank you for the insightful question. We agree that a full Adam-style convergence theory for DP-GRAPE would be valuable, and we view this as an important direction for future work. However, the behavior of Adam and related adaptive optimizers is still not fully understood theoretically; existing results typically require restrictive conditions, significant algorithmic modifications, or assumptions that do not align with our projection-based setup (and even worse in the DP setting). Extending such analyses to the private and projected setting would introduce an additional layer of complexity and is far beyond the scope of this work. In fact, existing DP optimization methods — such as DP-SGD-JL, Book-Keeping, Ghost Clipping, and DPZero/DP-ZO – either provide no convergence analysis at all or offer guarantees only for SGD-style algorithms.
>
> We would like to emphasize, however, that our SGD-style analysis remains directly relevant. In our theoretical development, the learning rate is kept constant, and under zero-momentum constraints, Adam essentially reduces to SGD. Thus, our convergence analysis for projected SGD can be interpreted as a valid upper bound on the performance of DP-GRAPE when used with Adam: the adaptive steps of Adam would only help, and our results provide a conservative theoretical guarantee that applies to this special case.

---

### Official Review · Reviewer_3fv9 · 2025-11-05

**Soundness:** 3
**Presentation:** 3
**Contribution:** 3
**Rating:** 6
**Confidence:** 3

**Summary:**

The authors propose projecting per-sample gradients onto low-dimensional random Gaussian subspaces before privatization, thus reducing memory and optimizer state size while maintaining DP guarantees.
The work is motivated by an empirical observation that differential privatization flattens the singular value spectrum of gradients, making SVD-based projections (e.g., GaLore) unnecessary. DP-GRAPE instead uses random Gaussian projections computed on-the-fly

**Strengths:**

The observation about spectral flattening is novel and provides a principled reason to abandon SVD-based projections.

The authors provide a theoretical privacy and convergence analysis for DP-GRAPE, which is non-trivial due to the introduction of random projections.

Evaluations cover both CV (ViT pre-training) and NLP (RoBERTa, OPT). Achieves large-scale DP training (OPT, 6.7B).

Memory savings in training are considerable: it cuts memory by over 63% in Vision Transformer training and 70% in RoBERTa fine-tuning compared to DP-Adam.

**Weaknesses:**

The privacy guarantee under random projections with unbounded entries is described informally. A more rigorous sensitivity or RDP proof sketch is needed.

DP-GRAPE’s algorithm is more complex to implement than vanilla DP-SGD/DP-Adam. I'm not sure how practical would be to implement it. No code mentioning.

**Questions:**

Have you done any ablation of projection dimension r versus accuracy/privacy?

How does the projection dimension r influence the effective privacy budget?

How does DP-GRAPE interact with existing memory-saving techniques like ghost clipping or even simple gradient accumulation?

DP-MERF, Harder at al. uses random features to create embeddings. It does not do it with memory efficiency as a goal but does not do it as a by-product?

---

> ### Author Response · Authors · 2025-11-23
>
> We thank the reviewer for the time taken to read our paper and for the constructive and thoughtful feedback. Please find our detailed responses below.
>
> > W1: The privacy guarantee under random projections with unbounded entries is described informally. A more rigorous sensitivity or RDP proof sketch is needed.
>
> We would like to clarify why the privacy analysis remains rigorous despite the use of projection matrices with unbounded entries. In DP-GRAPE, we clip gradients after projecting them. Our privacy analysis assumes that **the projection matrix is known to the adversary**; conditioned on this matrix, the projected gradients are simply a deterministic transformation of the original gradients. After projection, we deterministically clip the projected vectors and apply the moments accountant to obtain the differential privacy guarantees. Therefore, since the projection matrix is assumed public, the transformation is simply post processing and the differential privacy guarantees remain. As a result, the object being privatized is simply a sum of vectors with bounded norms, exactly paralleling the setting of DP-SGD.
>
> The same reasoning applies to Rényi Differential Privacy: since the mechanism reduces to applying Gaussian noise to clipped, subsampled gradients, its RDP parameters are exactly those of the well-known subsampled Gaussian mechanism [1]. The projection step does not alter either the sensitivity or the RDP computation as we assume that the projection matrix is known to the adversary. We will make this more clear in our proof sketch.
>
> [1] Mironov, I., Talwar, K., & Zhang, L. (2019). Rényi differential privacy of the sampled gaussian mechanism. arXiv preprint arXiv:1908.10530.
>
> > W2: DP-GRAPE’s algorithm is more complex to implement than vanilla DP-SGD/DP-Adam. I'm not sure how practical it would be to implement it. No code mentioning.
>
> We would like to emphasize that a complete and runnable implementation of DP-GRAPE is provided in the supplementary material. The algorithm is also practical to implement in standard deep-learning frameworks: all required components (projection operations, low-dimensional state updates, and seed-based reconstruction) are directly supported by PyTorch’s linear-algebra and random-number generator modules. In practice, we implemented DP-GRAPE on top of the existing Opacus library for differentially private training in PyTorch [2].
>
> [2] Yousefpour, A., et al. (2021). Opacus: User-friendly differential privacy library in PyTorch. arXiv preprint arXiv:2109.12298.
>
> > Q1: Have you done any ablation of projection dimension r versus accuracy/privacy?
>
> Yes, we conducted an ablation study on the effect of the projection dimension $r$ on both accuracy and privacy. In Figure 4 of Appendix C.2 in the revised upload, we provide additional plots showing RoBERTa fine-tuning accuracy on SST-2 for $r = 4, 8, 16, 32, 64, 128, 256$, using the same setup as in the experiments for Table 3. The results indicate that reducing $r$ below $32$ yields only marginal additional memory savings (as also shown in Figure 4), while the highest SST-2 accuracy is obtained at $r = 16$ for both tested privacy levels. Based on these observations, we recommend choosing $r \in [8, 32]$ for similar fine-tuning tasks.
>
> > Q2: How does the projection dimension r influence the effective privacy budget?
>
> We thank the reviewer for raising this question. In our setting, the projection dimension
> $r$ does not influence the effective privacy budget. Our privacy analysis assumes that the projection matrix is known to the adversary; conditioned on this matrix, the projected gradients are simply a deterministic transformation of the original gradients. After projection, we deterministically clip the projected vectors and apply the moments accountant to obtain the theoretical privacy guarantees. Therefore, since the projection matrix is assumed public, the transformation is simply post processing and the choice of r does not affect the privacy budget. This is an important point and we will add this in the paper.

---

> ### Author Response · Authors · 2025-11-23
>
> > Q3: How does DP-GRAPE interact with existing memory-saving techniques like ghost clipping or even simple gradient accumulation?
>
> Great question! Gradient accumulation is easily incorporated into the DP-GRAPE algorithm by computing partial sums of the projected and clipped sample gradients (step 12 of Algorithm 1) until the entire batch is completed. We used gradient accumulation with DP-GRAPE for many of our experiments, including the ViT pre-training and the OPT experiments. On the other hand, techniques such as bookkeeping and ghost clipping can also be integrated into DP-GRAPE. Notice that these methods only modify how per-sample gradients are computed and privatized; they do not alter the underlying optimization algorithm. Once the model gradients are computed using such optimized techniques, we can project them into a lower-dimensional space. After this projection, any additional privatization steps can be applied directly to the projected vectors, since our privacy analysis already assumes knowledge of the projection matrix. As a result, incorporating ghost clipping and bookkeeping into DP-GRAPE would not change the utility–privacy trade-off. In fact, using ghost clipping can further reduce memory usage for storing per-sample gradients, at the cost of a modest increase in per-iteration runtime. We have mentioned this in Page 9, lines 477 - 481.
>
> However, we would like to point out that to enable book-keeping and ghost clipping in our algorithm, we would need to clip using an upper bound on the projected gradient rather than the actual norm of the projected gradient. More concretely, if $M$ is a random matrix and $G$ is the per-sample gradient and $M^T G$ is the projected gradient update (we can concatenate all the layer-wise gradient matrices and corresponding random matrices), then we would need to clip by comparing $C$ with $|M|_F$ $|G|_F$  (where $|H|_F$ represents the Frobenius norm of a matrix)  instead of using the base projected gradient norm. This discrepancy can be corrected by increasing the learning rate by a factor that scales approximately with the square root of the total dimension random matrices divided by the per-entry standard deviation of the individual gaussian entries (at least in SGD) as we know that the value of ${|M|_F}^2$ would be tightly concentrated around its mean. Note that because we use the same projection matrix for every sample, we only need to compute the norm of the projection matrix once, and then we can compute the per-sample norms $|G|_F$ using techniques such as book-keeping and ghost clipping. Our convergence analysis does not change under this setting, as our high-probability event assumes that our upper bound is less than or equal to $C$ (and sets $C$ accordingly), rather than relying on concentration of the actual projected gradient norm. We will add a brief clarification on this in the final version of our paper.
>
> > Q4: DP-MERF, Harder at al. uses random features to create embeddings. It does not do it with memory efficiency as a goal but does not do it as a by-product?
>
> We would first like to respectfully clarify that DP-MERF (Harder et al.) is a method developed specifically for DP synthetic data generation, which is fundamentally different from the DP optimization and fine-tuning tasks considered in our work. While DP-MERF employs random features to construct kernel embeddings, memory efficiency is neither its intended objective nor a natural by-product in the context relevant to optimization.
>
> However, if one were to adopt their idea directly (namely, to perform DP optimization on a random projection of the input data) this would typically introduce notable limitations. Such an approach can degrade utility in general or require additional assumptions on the input data distribution as well as restrictions on the function class. By contrast, our method provides convergence guarantees for a significantly broader set of function classes without imposing assumptions on the underlying data.
>
> We would also like to emphasize that feature-level compression by itself does not necessarily lead to memory savings during training. Unless the model architecture is changed to reflect the reduced feature dimension, the memory required for optimization (which is dominated by the model parameters and associated optimizer states) remains essentially unchanged (aside from a small change to the first layer). Because our approach modifies the optimization procedure rather than the model architecture, we obtain substantial memory savings for any given model, whereas feature-level compression alone would not provide this benefit.
> In summary, although DP-MERF uses random-feature embeddings for a different purpose, directly extending their idea to DP optimization would reduce utility, impose stronger assumptions, and would not, on its own, offer meaningful memory savings. We will clarify this point in our revision.

---

### Official Review · Reviewer_uZQ1 · 2025-11-07

**Soundness:** 3
**Presentation:** 3
**Contribution:** 3
**Rating:** 6
**Confidence:** 4

**Summary:**

This work presents a new approach to memory-efficient DP training using random projections instead of SVD-based subspaces, which is motivated by a “flattened” singular value spectrum after privatization. DP-GRAPE (Gradient RAndom ProjEction) employs three key components: (1) random Gaussian matrices replace SVD-based subspaces, (2) gradients are privatized after projection, and (3) projection is applied during  backpropagation. The experiments show that DP-GRAPE can reduce the memory footprint of DP training without sacrificing accuracy or training time.

**Strengths:**

- using random projections (DP-GRAPE) instead of SVD-based projections, which is memory efficient.
- DP-GRAPE (Gradient RAndom ProjEction) achieves a privacy-utility trade-off comparable to DP-SGD.
- The margins in the experiments are significant, in terms of the memory reduction, while preserving the accuracy.

**Weaknesses:**

- Comparisons asre not sufficient with SOTA methods, and other subspace methods.
- The robustness analysis for failure cases is missing.

**Questions:**

- The differences between the DP-GRAPE and existing subspace methods, such as LoRA, etc.
- The robustness analysis for failure cases is missing.
- Hyperparameters, 'somewhat extensive hyperparameter searches', sensitivity analysis is necessary.

---

> ### Author Response · Authors · 2025-11-23
>
> We thank the reviewer for the time taken to read our paper and for the constructive and thoughtful feedback. Please find our detailed responses below.
>
> > W1: Comparisons are not sufficient with SOTA methods, and other subspace methods.
>
> Thank you for raising this point about comparisons with other subspace-based methods. To the best of our knowledge, the main techniques in this category (both in private and non-private optimization) are LoRA and GaLore. We included a direct comparison with DP-LoRA in Table 3 (Page 8), and we also report results for a naive DP variant of GaLore in Table 9 (Page 16). These baselines represent the subspace methods most closely related to our setting, and we hope the added clarity helps contextualize our contributions.
>
> > W2: The robustness analysis for failure cases is missing.
>
>  Can you please clarify which specific types of failure cases you are referring to? If there are particular statistics or robustness metrics you would like us to report, we would be happy to include them in a revised version.
>
> > Q1: The differences between the DP-GRAPE and existing subspace methods, such as LoRA, etc.
>
> The key difference between DP-GRAPE and existing subspace methods such as LoRA lies in what is being modified. LoRA changes the model architecture by adding trainable low-rank matrices while keeping the original weights frozen. Its DP version, DP-LoRA, simply applies DP-SGD to these low-rank parameters, meaning the underlying optimization algorithm stays the same and the privacy guarantee comes directly from DP-SGD. While this works well for fine-tuning, the low-rank assumption is too restrictive for pre-training, which is why LoRA generally cannot be used in that setting.
>
> In contrast, DP-GRAPE modifies the optimization algorithm rather than the model architecture. We change the gradient estimator itself by projecting gradients into a random low-dimensional subspace before privatization, making the method a variant of a stochastic-gradient algorithm with a different estimator. This design allows DP-GRAPE to perform well in both fine-tuning and pre-training.
>
> Regarding GaLore, the distinction is different. We discuss the differences between DP-GRAPE and a naive DP adaptation of GaLore on Page 3, lines 169–179. Our motivation here comes from observing that simply adding DP noise and clipping to existing subspace methods like GaLore eliminates much of their benefit: the clipping and noise flatten the gradient spectrum, making the low-rank structure less meaningful. Under DP noise, random projections maintain their effectiveness and can outperform such naive DP subspace methods while providing significant improvements.
>
> > Q2: Hyperparameters, 'somewhat extensive hyperparameter searches', sensitivity analysis is necessary.
>
> We address hyperparameter selection and tuning in detail in Appendix C. In particular, we describe our hyperparameter search procedures (based on grid search) in the first two paragraphs of Appendix C.1 (Vision Transformer experiments), C.2 (RoBERTa experiments), and C.3 (OPT experiments). We additionally provide practical hyperparameter recommendations for training these models in Appendix C.4.
> Regarding sensitivity analysis, we report the sensitivity calculations corresponding to the add-one/remove-one notion of differential privacy on Page 22, lines 1163–1168. We hope these details help address the reviewer’s concerns.

---

### Meta-Review · Area_Chair_mm2L · 2025-12-22

**Summary:**

This paper introduces DP-GRAPE, a memory-efficient method for differentially private training that uses random low-rank projections of gradients. While reviewers recognize the significant practical potential of the work, particularly its novel motivation (spectral flattening under DP noise), strong theoretical analysis, and impressive memory reduction (up to ~70%) on billion-parameter models, the consensus points to critical missing elements that prevent acceptance in its current form. The most significant weaknesses are incomplete empirical validation and positioning. The paper lacks direct, head-to-head comparisons with key state-of-the-art memory-efficient DP methods (e.g., DP-SGD-JL, Ghost Clipping, Book-Keeping, DP-AdamBC, DP-ZO). This omission leaves the claims in Table 1 unsupported and makes it impossible to assess the true contribution relative to the current frontier. Furthermore, the privacy analysis for unbounded random projections is described informally and requires a more rigorous proof sketch. Other major concerns include the lack of a detailed memory breakdown (parameters vs. gradients vs. optimizer states) and insufficient analysis of hyperparameter sensitivity (especially the projection dimension). The core idea is valuable and the results are promising, but the paper is not yet complete. To meet the bar for publication, the authors must: (1) Conduct comprehensive comparisons with the missing SOTA baselines under identical settings; (2) Provide a more rigorous privacy analysis for the random projection mechanism; (3) Include a detailed memory attribution analysis and a thorough ablation study on the key hyperparameters.

These revisions are substantial and fundamental to evaluating the paper's contribution; they cannot be addressed through minor edits.

**Reviewer Concerns:**

I think the concerns from Reviewer gYpZ have been addressed. However, concerns from other reviewers are still outstanding, though some of them have been clarified through the rebuttal.

**Reviewer Scores:**

Reviewer gYpZ has raised the score, but the rest of the reviewers will maintain the scores.

---

### Decision · Program_Chairs · 2026-01-26

Reject